# Covariant Lyapunov Vectors and Finite-Time Normal Modes for Geophysical Fluid Dynamical Systems

**DOI:** 10.3390/e25020244

**Published:** 2023-01-29

**Authors:** Jorgen S. Frederiksen

**Affiliations:** CSIRO Environment, Aspendale, Melbourne 3195, Australia; jorgen.frederiksen@csiro.au

**Keywords:** Lyapunov vectors, singular vectors, Floquet vectors, finite-time normal modes, Oseledec theorem, entropy production, degeneracy, geophysical fluid dynamics, chaotic dynamics

## Abstract

Dynamical vectors characterizing instability and applicable as ensemble perturbations for prediction with geophysical fluid dynamical models are analysed. The relationships between covariant Lyapunov vectors (CLVs), orthonormal Lyapunov vectors (OLVs), singular vectors (SVs), Floquet vectors and finite-time normal modes (FTNMs) are examined for periodic and aperiodic systems. In the phase-space of FTNM coefficients, SVs are shown to equate with unit norm FTNMs at critical times. In the long-time limit, when SVs approach OLVs, the Oseledec theorem and the relationships between OLVs and CLVs are used to connect CLVs to FTNMs in this phase-space. The covariant properties of both the CLVs, and the FTNMs, together with their phase-space independence, and the norm independence of global Lyapunov exponents and FTNM growth rates, are used to establish their asymptotic convergence. Conditions on the dynamical systems for the validity of these results, particularly ergodicity, boundedness and non-singular FTNM characteristic matrix and propagator, are documented. The findings are deduced for systems with nondegenerate OLVs, and, as well, with degenerate Lyapunov spectrum as is the rule in the presence of waves such as Rossby waves. Efficient numerical methods for the calculation of leading CLVs are proposed. Norm independent finite-time versions of the Kolmogorov-Sinai entropy production and Kaplan-Yorke dimension are presented.

## 1. Introduction

Instability, error growth, and entropy production are fundamental interrelated properties of the dynamics and predictability of chaotic systems. Canonical equilibrium states of maximum entropy are also nonlinearly stable states in the high-resolution limit when fluctuations vanish [1,2,3,4,5]. More generally, the Shannon entropy [6,7] can be used to establish the Kolmogorov-Sinai (KS) entropy production [8,9] that quantifies the chaotic nature of dynamical systems. A simple plausible measure of KS entropy production is given by Pesin’s formula [10] that expresses it as the sum of the positive Lyapunov exponents [11]. These exponents describe the long-term average growth rates of linear instabilities evolving on the time-varying flows. Associated local and finite-time analogues of KS entropy production have also been proposed by Wei [12] and Quinn et al. [13].

In the case of geophysical fluid dynamical systems, linear normal mode baroclinic instability theory, with simple stationary zonally symmetric basic states, was used by Charney, Eady, and Phillips [14,15,16] to explain the basic mechanism of extratropical storm formation. Frederiksen [17] developed a theory of localized cyclogenesis that explains the structures of storm tracks based on the instability of three-dimensional climatological basic states. Indeed, as reviewed in [18], three-dimensional instability theory yields analogues of essentially all the major large-scale atmospheric disturbances in both hemispheres including storms [19,20], blocks [19,20], teleconnection patterns [20,21,22,23], intraseasonal oscillations [24] and the classes of convectively coupled tropical disturbances [25]. These disturbances are largely propagating modes with complex conjugate eigenvalues and eigenvectors apart from some stationary teleconnection patterns.

Three-dimensional instability theory also sheds light on the local growth of errors with basic states that are snapshots of the flow fields [26]. Indeed, as shown by Wei and Frederiksen [27], leading normal modes of local flow fields are reasonably successful in capturing the structures of error growth over the subsequent couple of days. In general, however, leading finite-time dynamical vectors provide more accurate predictors of the structures and amplitudes of evolved errors.

A major motivation for studying the properties of finite-time dynamical vectors has been to understand error growth in weather prediction, and subsequently seasonal prediction, and for ameliorating the effect of errors through ensemble prediction. Lorenz [28] first considered the growth of ensembles of small errors in a low order atmospheric model and showed it was related to the singular values of the tangent linear propagator. Finite-time and local singular values, exponents, and singular vectors (SVs) were subsequently employed in many dynamical and predictability studies with relatively simple and intermediate complexity models [13,26,29,30,31,32,33,34,35,36] as reviewed in [13,36].

In some studies finite-time singular value exponents are called finite-time Lyapunov exponents since they converge to the global Lyapunov exponents in the long-time limit. Indeed, since the 1990s Lyapunov exponents and orthonormal Lyapunov vectors (OLVs) have frequently been used in the analysis of predictability within simple and intermediate complexity models of geophysical flows [35,37,38,39,40,41,42], as reviewed in [41,42]. Increasingly, covariant Lyapunov vectors (CLVs) have also gained popularity [13] as will be discussed in more detail below.

Toth and Kalnay [43,44] introduced a simple breeding method for ensemble perturbation generation in a comprehensive weather forecasting model at National Centers for Environmental Prediction (NCEP) and likened the bred vectors to stochastically and nonlinearly modified leading Lyapunov vectors. These perturbations help ameliorate the effects of fast-growing instabilities that cause rapid loss of predictability particularly during flow regime transitions such as into and out of blocking states [45].

At the European Centre for Medium Range Forecasting (ECMWF), Molteni et al. [46] used ensemble perturbations consisting of mixtures of finite-time right (initial) and left (evolved) SVs of the propagator. The extensive development of the SV method for ensemble perturbations is reviewed by Leutbecher and Palmer [47] and Quinn et al. [13]. The SV approach can induce non-modal perturbation growth even in overdamped systems [13]. Wei and Toth [48] found both the bred vector and SV schemes had similar performance in improving weather forecasts.

Frederiksen [49,50] proposed finite-time normal modes (FTNMs) of the propagator as ensemble perturbations. FTNMs can be defined for any time-period of interest between an initial time $t_{0}$ and final time $t_{f}$. They are eigenmodes of the propagator $G(t_{f},t_{0})\phi (t_{0})=\lambda \phi (t_{0})=\phi (t_{f})$ where G is the propagator, ϕ is a FTNM and λ is the eigenvalue. Because of the eigenvalue relationship $\phi (t_{f})=\lambda \phi (t_{0})$ FTNMs have the remarkable property of having norm independent growth rate ${\left(t_{f}-t_{0}\right)}^{-1}\ln\left|\lambda \right|$ between $t_{0}$ and $t_{f}$ as well as norm independent structures. They are also covariant with the tangent linear dynamics for $t_{f}\geq t\geq t_{0}$ with $\phi (t)=G(t,t_{0})\phi (t_{0})$.

Wei and Frederiksen [27,41] found that leading FTNMs were better predictors of evolved error structure and amplitude than leading SVs, OLVs and local normal modes in barotropic forecasts perturbed by initial random errors. This is the case particularly over shorter time scales with the performance of leading Lyapunov vectors approaching that of FTNMs after about 14 days in tangent linear integrations [41]. It should be noted, however, that after about 3 days nonlinear effects start to affect error growth of atmospheric synoptic scale disturbances [51].

Interestingly, in coupled ocean-atmosphere seasonal hindcasts Frederiksen et al. [52] found that monthly optimized cyclic modes, essentially stochastically and nonlinear modified leading FTNMs, were much better ensemble perturbations than bred vectors [53]. Sandery and O’Kane [54] also found that cyclic modes were effective perturbations for the initialization and ensemble prediction in an ocean-atmosphere tropical cyclone prediction system. These results are of course consistent with the fact that local and finite-time instabilities and growth rates are closer related to shorter time error growth and predictability than global instabilities and exponents [12,13,27,30,31,32,33,34,35,37,41].

FTNMs for a single time step reduce to normal modes and become Floquet vectors [55] if the flow is periodic, so that the initial and final basic states are the same. Frederiksen [50] found that the different structures of initial SVs, depending on field variables or norm, and their super exponential growth could be explained by their projection onto FTNMs. Super exponential growth of SVs is largely explained by their large projection onto the leading FTNMs and the interference effects of subdominant FTNMs reducing the norm to say unity. With time the interference effects of the subdominant FTNMs disappear leaving the leading FTNM with large magnitude.

Wolfe and Samelson [56] have also emphasized the importance of norm-independence of dynamical vectors used to understand error growth and predictability. Their focus, like that of Ginelli et al. [57] has been to implement efficient algorithms for calculating covariant Lyapunov vectors (CLVs) for aperiodic systems from long-time SVs that approximate OLVs. These algorithms and subsequent variations [58,59,60] have allowed the practical calculation of CLVs for nondegenerate systems. CLVs have norm independent structures like normal modes, Floquet vectors and FTNMs. They are covariant with the tangent linear dynamics over the whole time-domain. However, their finite-time growth rates are norm dependent. The general properties of CLVs were discussed in the earlier works of Ruelle [61], Eckman and Ruelle [62], Vastano and Moser [63], Legras and Vautard [64] and Trevisan and Pancotti [65].

The development of more efficient algorithms for the calculation of CLVs [56,57,58,59,60] for aperiodic systems has led to a flurry of activity in applications and further theoretical developments much of which has been reviewed by Quinn et al. [13]. For geophysical fluid dynamical systems, of primary interest here, although our results apply more generally, there have been developments and applications of CLVs in studies of the stability of atmospheric flows [66,67], of atmospheric blocking [68] and large-scale low frequency teleconnection patterns [13,69] and of predictability and data assimilation in coupled ocean-atmosphere systems [70,71].

In this article we make a detailed analysis of the relationships between CLVs, OLVs, SVs and Floquet vectors and FTNMs. The main goals are:To establish the relationships between long-time FTNMs and CLVs, and their growth rates, for aperiodic, as well as periodic, chaotic dynamical systems;To deduce these properties for systems with degenerate as well as nondegenerate Lyapunov spectra;To examine and propose methods for the calculation of CLVs and Lyapunov exponents that allow for degenerate Lyapunov spectra.

The article is organized in two parts. In the first part, (Section 2, Section 3, Section 4, Section 5, Section 6 and Section 7, Appendix B and Appendix D) the properties of the dynamical vectors are summarized for dynamics in general phase-spaces. A notation is used that is suggestive of the relationships between the norm dependent orthonormal OLVs (𝓾,𝓿) and SVs (u,v), and the norm independent nonorthogonal CLVs (ψ) and FTNMs (ϕ), represented by Greek symbols, (and associated subspaces and characteristic matrices). The eigenvalues or amplification factors (λ,σ,ℓ) are in lower case and exponents or growth rates (Λ,Σ,L) in corresponding capitals. In the second part (Section 8, Section 9, Section 10 and Section 11.4, Appendix A, Section B.5, Appendix C and Appendix E) the simplifications and connections between SVs, OLVs, CLVs and FTNMs and their exponents that occur when the tangent linear dynamics are analysed in FTNM-space are established. These findings are then used to deduce the three main aims, 1 to 3 above, as well as other results.

In detail, the structure of the article is as follows: The tangent linear equations for smooth ergodic dynamical systems with bounded attractors are summarised in Section 2 where the propagator and its semi-group or cocycle properties are also documented. The FTNMs are defined in Section 3 as the eigenvectors of the propagator and their attributes as well as those of the eigenvalues and FTNM exponents presented. The finite-time adjoint modes are also described in this Section. Floquet vectors for periodic systems and their relationships to FTNMs are discussed in Section 4, and in Section 5, SVs and singular values and exponents are considered. Section 6 contains a brief outline of Lyapunov vectors and exponents and the Oseledec (also known as Oseledets) multiplicative ergodic theorem [72,73,74,75] that governs their behaviour, with recent advances established and reviewed in [74,75]. The Gram-Smidt method for the construction of all OLVs from CLVs and the inverse method for the construction of all CLVs from OLVs are presented initially for nondegenerate OLVs in Section 7. There more efficient methods for calculation of just some of the leading CLVs from OLVs [56,57,58,59,60] are also mentioned. The case of degenerate Lyapunov spectrum is also discussed.

In Section 8, the transformation of the dynamical equations, and particularly the tangent linear equations and propagator, into the phase-space of FTNM coefficients is performed. The eigenvalue-eigenvector equations for FTNMs and SVs are developed and the relationships between FTNMs and SVs and FTNM eigenvalues and singular values and exponents in this phase-space determined. The findings that, in FTNM-space, singular value exponents equal FTNM growth rates and SVs equate to unit norm FTNMs at critical times underpin the deductions in Section 9 and Section 10. For periodic systems, the anchoring of SVs with FTNMs in FTNM-space establishes the connection of FTNMs, through SVs, to OLVs and CLVs, from the results in Section 7, and the equivalence of Floquet vectors and CLVs, including for degenerate Lyapunov spectra. Aperiodic systems are then analysed in Section 10 where it is shown that long-time SVs, essentially OLVs, equate, at certain times, to long-time FTNMs in the phase-space of FTNM coefficients. This, together with the expression for the construction of CLVs from OLVs in Section 7, and the covariant properties of both CLVs and FTNMs establishes the equivalence of CLVs with long-time FTNMs. These results are shown to hold for both the case of nondegenerate and degenerate Lyapunov spectra.

In Section 11, current numerical methods for the calculation of leading CLVs, which have largely been applied in situations when the associated OLVs are nondegenerate, are briefly discussed. It is pointed out through specific examples that for geophysical fluid dynamical systems incorporating waves, such as Rossby waves, the SVs and OLVs are frequently degenerate and the related FTNMs, Floquet vectors and CLVs occur mainly in complex conjugate pairs. Efficient methods for the direct calculation of leading FTNMs, and Floquet vectors and CLVs as long-time FTNMs, are discussed. Norm independent versions of the Kolmogorov-Sinai entropy production and Kaplan-Yorke dimension are presented. In Section 12 the implications of the results deduced are discussed and summarised and conclusions drawn.

In Appendix A, the method of ordering the FTNM eigenvalues and eigenvectors when some of the eigenvalues are complex is presented. In Appendix B, the content of the Oseledec multiplicative ergodic theorem that governs the tangent linear dynamics is summarised. This includes the Oseledec operators and their eigenvectors and associated Lyapunov exponents, and the Oseledec subspaces in both the nondegenerate and degenerate cases. Here, the simplification that occurs when the phase-space is FTNM-space is also presented. Appendix C considers Lyapunov homologous transformations of the propagator cocycle between FTNM phase-space and general phase-spaces. The equivalence of CLVs in different phase-spaces, or their norm independence, and that of the global Lyapunov exponents, are summarised in Appendix D. There the norm dependence of finite-time Lyapunov exponents is also recapped. In Appendix E, an alternative method to that of Section 10, of relating long-time FTNMs and Lyapunov vectors is developed that leads to the same conclusion that long-time FTNMs converge to CLVs under the specified conditions on the dynamical systems.

## 2. Dynamical Equations

In this study, we consider smooth ergodic dynamical systems with bounded attractors that generate well defined statistics [30,72,76,77]. We write the nonlinear evolution equations for vector fields X(t) in the form
(1)dX(t)dt=N(X(t))
where N(X(t)) is a smooth nonlinear matrix operator. Here X=(X1,X2,…,XN)T is the column vector of dynamical fields with N components and T denotes transpose. We suppose that X(t) is perturbed by a disturbance x(t) which is sufficiently small that it satisfies a linearized equation about the trajectory X(t) for the time interval of interest. Linearization of Equation (1) about the trajectory X(t) yields the tangent linear equation for the perturbation x(t)=(x1(t),x2(t),…,xN(t))T:(2)dx(t)dt=M(t)x(t)
where the Jacobian dynamical matrix M(t)≡M(t;{X(t)}), taken to be nonsingular, depends on the trajectory X(t). In physical space the vector fields belong to the real N-dimensional space ℝN. It is often convenient to perform associated or intermediary dynamical and instability calculations and their analysis in other spaces and then transform back to physical space [72] (see also Appendix C). For example, calculations are often performed in Fourier [1] or spherical harmonic space [3,17], and in empirical orthogonal function space [71]. The results in [50] where SVs were projected in terms of FTNMs also suggests that analyzing the dynamics in FTNM-space would be insightful. For this reason, in the following analysis we formulate definitions of the dynamical matrices and their relationships in ways where they also apply to complex fields. For example, we employ the Hermitian conjugate denoted by superscript † which reduces to the transpose T for real matrices. Of course, one can always revert to the real domain by writing the complex expressions in terms of their real and imaginary parts [61,78] (see also Appendix C) or directly transform back to physical space; however, the complex representation may sometimes be more convenient or elegant.

The solution of Equation (2) is
(3)$$x(t)=G(t,t_{0})x(t_{0})$$
where
(4)$$G(t,t_{0})\equiv G(t,t_{0};\{X(t)\leftarrow X(t_{0})\}).$$Here, $G(t,t_{0})$ is the propagator or cocycle from the initial time $t_{0}$ to time t that depends on the whole trajectory between these times denoted by $\left\{X(t)\leftarrow X(t_{0})\right\}$. We use the notation where the time flows from right to left in the propagators as for the standard retarded propagators [49].

We see from Equations (2) and (3) that $G(t,t_{0})$ satisfies the differential equation
(5)$$\frac{dG(t,t_{0})}{dt}=M(t)G(t,t_{0}).$$The formal solution to Equation (5) is
(6)$$G(t,t_{0})=T\exp\left[\int_{t_{0}}^{t}dsM(s)\right]$$
where T is the chronological time-ordering operator [30,49]. Here $G(t,t_{0})$ satisfies the semi-group or multiplicative cocycle properties
(7)$$G(t,t_{0})=G(t,\tau )G(\tau ,t_{0})\;;\;G(t_{0},t_{0})=I$$
and the propagators are assumed to be nonsingular with inverse ${\left[G(t,t_{0})\right]}^{-1}$ describing the reverse propagation from t to $t_{0}$ as in Oseledec [72]. Here, I=diag(1,1,…,1) is the unit matrix with diagonal elements of 1 and the off-diagonal elements are zero.

For simple and intermediate complexity systems [79,80], the propagator may be calculated by using its cocycle properties
(8)$$G(t,t_{0})=G(t,t_{j-1})G(t_{j-1},t_{j-2})\ldots G(t_{2},t_{1})G(t_{1},t_{0})$$
as in Refs. [72] and [80], Equation (2.8b) or as described in [49]. Here $t=t_{j}$ and the constituent propagators are for short time steps $\delta t=t_{k}-t_{k-1}$ between $t_{k-1}$ and $t_{k}$. Using a predictor-corrector time step the short-time propagators take the form
(9)$$G(t_{k},t_{k-1})=I+\delta tM\left(\frac{t_{k}+t_{k-1}}{2}\right)+\frac{1}{2}{\left(\delta t\right)}^{2}M^{2}\left(\frac{t_{k}+t_{k-1}}{2}\right)$$
as shown in [80], Equation (2.8a). This is the second order truncation of the associated exponential form as in Equation (6). For more complex models, the leading FTNMs can be obtained using the method of Wei and Frederiksen [27] (Appendix) that is discussed in Section 11.3.

## 3. Eigenvectors and Eigenvalues of the Propagator

Next, we examine the eigenvectors of the propagator, termed finite-time normal modes (FTNMs), and the associated adjoint modes [49], as well as their eigenvalues.

### 3.1. Finite-Time Normal Modes

In this study, we consider the dynamical equations, the propagator and eigenvalues and eigenvectors of a number of matrices over various time intervals T between $t_{0}$ and $t_{f}$. Here,
(10)$$T=T(t_{f},t_{0})=[t_{f}-t_{0}]=t_{f}-t_{0},$$
and $t_{f}>t_{0}$ so that T>0. It is convenient to represent the dependence of these quantities on different optimization time intervals by the notation $\left[t_{f}-t_{0}\right]$. This is also a reminder that time flows from right to left in the propagators.

The FTNMs are the eigenvectors of the propagator
(11)$$G(t_{f},t_{0})\phi^{n}(t_{0};[t_{f}-t_{0}])=\lambda^{n}(t_{f},t_{0})\phi^{n}(t_{0};[t_{f}-t_{0}])=\phi^{n}(t_{f};[t_{f}-t_{0}])$$
where the eigenvalues are ordered so that
(12)$$\left|\lambda^{1}\right|\geq \left|\lambda^{2}\right|\geq \ldots \geq \left|\lambda^{N}\right|$$
and n=1,2,…,N. The in general complex eigenvalue $\lambda^{n}(t_{f},t_{0})$ can be written in terms of the real and imaginary parts as
(13)$$\lambda^{n}(t_{f},t_{0})=\lambda_{r}^{n}+i\lambda_{i}^{n}=\exp\Lambda^{n}(t_{f},t_{0})(t_{f}-t_{0})=\exp\{\Lambda_{r}^{n}+i\Lambda_{i}^{n}\}(t_{f}-t_{0}).$$Here,
(14)$$\Lambda_{r}^{n}(t_{f},t_{0})={\left(t_{f}-t_{0}\right)}^{-1}\ln\left|\lambda^{n}\right|=T^{-1}\ln\left|\lambda^{n}\right|$$
and
(15)$$\Lambda_{i}^{n}(t_{f},t_{0})={\left(t_{f}-t_{0}\right)}^{-1}\arctan\left[\frac{\lambda_{i}^{n}}{\lambda_{r}^{n}}\right]=T^{-1}\arctan\left[\frac{\lambda_{i}^{n}}{\lambda_{r}^{n}}\right].$$From Equation (12), the real parts of the exponents are ordered such that
(16)$$\Lambda_{r}^{1}\geq \Lambda_{r}^{2}\geq \ldots \geq \Lambda_{r}^{N}.$$We note that $\Lambda_{r}^{n}(t_{f},t_{0})$ is the average growth rate and $-\Lambda_{i}^{n}(t_{f},t_{0})$ is the average phase frequency [49], Equation (4.5). We assume that the possibly complex $\lambda^{n}$ are distinct and the associated eigenvectors are nondegenerate, which appears to be the generic case based on our previous studies [27,41,49,50,79,80]. In this case, when some of the eigenvalues are complex, we order them and the associated exponents and FTNM eigenvectors as described in Appendix A. In fact, our results would also be valid for different orderings that are consistent between the FTNMs and other dynamical vectors considered in this article.

The FTNMs have the important property of being norm independent, unlike SVs, and, when nondegenerate, can be used as a basis for representing any initial vector and its evolution. Thus, with
(17)$$x(t_{0})=\sum_{n=1}^{N}\kappa_{n}\phi^{n}(t_{0})$$
where $\kappa_{n}$ are the expansion coefficients in terms of FTNMs we find that
(18)$$x(t_{f})=\sum_{n=1}^{N}\lambda^{n}(t_{f},t_{0})\kappa_{n}\phi^{n}(t_{0}).$$Equation (18) illustrates another important property of FTNMs and that is that any initial disturbance is filtered by the dynamics in favor of the faster growing FTNMs with larger amplification factors $\left|\lambda^{n}\right|$ or growth rates $\Lambda_{r}^{n}$. The FTNMs for $t_{f}\geq t\geq t_{0}$ are defined by forward propagation through
(19)$$\phi^{n}(t;[t_{f}-t_{0}])=G(t,t_{0})\phi^{n}(t_{0};[t_{f}-t_{0}])$$
and hence the FTNMs are covariant with the dynamics over this interval.

The two definitions in Equations (11) and (19) only involve forward integrations of the dynamical equations or their associated propagators. However, based on these two definitions further properties of $\phi^{n}(t)$ may be established. Firstly, for $t_{f}\geq t\geq t_{0}$, $\phi^{n}(t;[t_{f}-t_{0}])$ satisfies the eigenvalue equation
(20)$$\lambda^{n}\phi^{n}(t)=G(t,t_{0})\lambda^{n}\phi^{n}(t_{0})=G(t,t_{0})\phi^{n}(t_{f})=G(t,t_{0})G(t_{f},t)\phi^{n}(t).$$Equivalently, taking time to be increasing through the recycling step (discussed in more detail in Section 11.3) such that t→t+T and $t_{0}\to t_{0}+T$ in the propagator $G(t,t_{0})$ we have
(21)$$G(t+T,t_{0}+T)G(t_{0}+T,t)\phi^{n}(t)=\lambda^{n}\phi^{n}(t).$$Here, $\lambda^{n}=\lambda^{n}(t_{f},t_{0})$ and
(22)$$G(t+T,t_{0}+T)\equiv G(t,t_{0}).$$Thus, one can also regard the FTNMs, $\phi^{n}(t)$, at time t as the solution of the eigenvalue-eigenvector equations in Equation (21) that involve a recycling of the disturbances. These relationships are consistent with the FTNMs being calculated by the algorithms that involve the recycling of initial perturbations as discussed in Section 11.3.

From Equation (11) we also see that
(23)$$\Phi (t_{f};[t_{f}-t_{0}])=G(t_{f},t_{0})\Phi (t_{0};[t_{f}-t_{0}]);\;G(t_{f},t_{0})=\Phi (t_{f};[t_{f}-t_{0}])\Phi^{-1}(t_{0};[t_{f}-t_{0}])$$
and
(24)$$G(t_{f},t_{0})\Phi (t_{0};[t_{f}-t_{0}])=\Phi (t_{0};[t_{f}-t_{0}])D_{\lambda }(t_{f},t_{0});\;G(t_{f},t_{0})=\Phi (t_{0};[t_{f}-t_{0}])D_{\lambda }(t_{f},t_{0})\Phi^{-1}(t_{0};[t_{f}-t_{0}])$$
with
(25)$$D_{\lambda }(t_{f},t_{0})=\mathrm{diag}(\lambda^{1},\lambda^{2},\ldots ,\lambda^{N}).$$Here, the characteristic matrix of the FTNMs
(26)$$\Phi (t;[t_{f}-t_{0}])=(\phi^{1}(t;[t_{f}-t_{0}]),\phi^{2}(t;[t_{f}-t_{0}]),\ldots ,\phi^{N}(t;[t_{f}-t_{0}])).$$We assume that the FTNMs are nondegenerate and that Φ(t) is nonsingular with the inverse $\Phi^{-1}(t)$ well defined. This then ensures that $G(t,t^{\prime })=\Phi (t)\Phi^{-1}(t^{\prime })$ is nonsingular with well-defined inverse ${\left[G(t,t^{\prime })\right]}^{-1}$ as in Oseledec [72].

### 3.2. Finite-Time Adjoint Modes

We consider next the adjoint equation corresponding to Equation (2)
(27)$$-\frac{da(t)}{dt}=M^{†}(t)a(t)$$
as in Ref. [49] where the propagator for backwards integration satisfies
(28)$$a(t_{0})=H(t_{0},t)a(t)$$
and
(29)$$H(t_{0},t)=G^{†}(t,t_{0}).$$Here, we also note that the Hermitian conjugate reverses the time integration as does the inverse operation.

From Equation (24) we note that
(30)$$G(t_{f},t_{0})=\Phi D_{\lambda }(t_{f},t_{0})\Phi^{-1}=\Phi D_{\lambda }(t_{f},t_{0})A^{†};\;G^{†}(t_{f},t_{0})=H(t_{0},t_{f})=AD_{\lambda^{*}}(t_{f},t_{0})\Phi^{†}$$
with
(31)$$A^{†}=\Phi^{-1}(t_{0};[t_{f}-t_{0}]).$$The matrix A consists of the columns of adjoint eigenvectors
(32)$$A=(\alpha^{1},\alpha^{2},\ldots ,\alpha^{N})$$
that are determined by
(33)$$G^{†}(t_{f},t_{0})\alpha^{n}=\lambda^{n*}\alpha^{n}$$
where ${}^{*}$ denotes complex conjugate. The FTNM eigenvectors and adjoint eigenvectors form a biorthogonal system with the adjoint eigenmodes normalized such that the Euclidean inner product
(34)$$\left(\alpha^{m},\phi^{n}\right)={\left(\alpha^{m}\right)}^{†}\phi^{n}=\delta_{mn}$$
with $\delta_{mn}$ the Kronecker delta function that is unity when m=n and zero otherwise.

## 4. Floquet Vectors

We consider here the case when the stability matrix M(t)≡M(t;{X(t)}) is periodic due to the periodicity of the trajectory X(t). Then the tangent linear equations for the dynamical systems considered in Section 2 satisfy the conditions for Floquet theory [55,65,79,80,81,82,83,84,85]. Thus, if the dynamical system is periodic so that
(35)M(t+T)=M(t)
then Equation (21), which applies to FTNMs, becomes
(36)$$G(t+T,t)\phi^{n}(t)=\lambda^{n}(t_{f},t_{0})\phi^{n}(t).$$For the periodic system the difference is that Equation (35) means that there is no discontinuity at $t_{f}=t_{0}+T$ and therefore
(37)$$G(t+T,t_{0}+T)G(t_{0}+T,t)=G(t+T,t).$$Moreover, we have
(38)$$G(t+KT,t)={\left[G(t+T,t)\right]}^{K}$$
where K is a positive integer and
(39)$$\begin{array}{l} G(t+KT,t_{0})\phi^{n}(t_{0};[t_{f}-t_{0}])=G(t+KT,t)G(t,t_{0})\phi^{n}(t_{0};[t_{f}-t_{0}])\\ =G(t+KT,t)\phi^{n}(t)={\left[\lambda^{n}(t_{f},t_{0})\right]}^{K}\phi^{n}(t). \end{array}$$Here, Equation (19) has been used for the last two equalities.

## 5. Singular Vectors

The propagator can also be presented through a singular value decomposition as
(40)$$G(t_{f},t_{0})=UD_{\sigma }V^{†}\;;\;G^{†}(t_{f},t_{0})=VD_{\sigma }U^{†}$$
with U and V unitary matrices and
(41)$$D_{\sigma }(t_{f},t_{0})=\mathrm{diag}(\sigma^{1}(t_{f},t_{0}),\sigma^{2}(t_{f},t_{0}),\ldots ,\sigma^{N}(t_{f},t_{0}))$$
is a diagonal matrix of singular values with $\sigma^{1}\geq \sigma^{2}\geq \ldots \geq \sigma^{N}$. The matrices U and V consist of the columns of the left SVs
(42)$$U=(u^{1},u^{2},\ldots ,u^{N})$$
and right SVs
(43)$$V=(v^{1},v^{2},\ldots ,v^{N}).$$Here, $u^{n}$ are the left orthonormal SVs that appear on the left in the first expression in Equation (40) and $v^{n}$ the right SVs. From Equation (40) we have
(44)$$G(t_{f},t_{0})v^{n}(t_{0};[t_{f}-t_{0}])=\sigma^{n}u^{n}(t_{f};[t_{f}-t_{0}])$$
and
(45)$$G^{†}(t_{f},t_{0})u^{n}(t_{f};[t_{f}-t_{0}])=H(t_{0},t_{f})u^{n}(t_{f};[t_{f}-t_{0}])=\sigma^{n}v^{n}(t_{0};[t_{f}-t_{0}]).$$Thus,
(46)$$G(t_{f},t_{0})G^{†}(t_{f},t_{0})u^{n}(t_{f};[t_{f}-t_{0}])=G(t_{f},t_{0})H(t_{0},t_{f})u^{n}(t_{f};[t_{f}-t_{0}])={\left(\sigma^{n}\right)}^{2}u^{n}(t_{f};[t_{f}-t_{0}])$$
and
(47)$$G^{†}(t_{f},t_{0})G(t_{f},t_{0})v^{n}(t_{0};[t_{f}-t_{0}])=H(t_{0},t_{f})G(t_{f},t_{0})v^{n}(t_{0};[t_{f}-t_{0}])={\left(\sigma^{n}\right)}^{2}v^{n}(t_{0};[t_{f}-t_{0}]).$$Equation (46) can also be written in the form:(48)$${\left[G^{†}(t_{f},t_{0})\right]}^{-1}{\left[G(t_{f},t_{0})\right]}^{-1}u^{n}(t_{f};[t_{f}-t_{0}])={\left(\sigma^{n}(t_{f},t_{0})\right)}^{-2}u^{n}(t_{f};[t_{f}-t_{0}]).$$The singular values and exponents are related through
(49)$$\sigma^{n}=\sigma^{n}(t_{f},t_{0})=\exp\sum^{n}(t_{f},t_{0})(t_{f}-t_{0})$$
with
(50)$$\sum^{n}(t_{f},t_{0})={\left(t_{f}-t_{0}\right)}^{-1}\ln(\sigma^{n}(t_{f},t_{0}))=T^{-1}\ln(\sigma^{n}(t_{f},t_{0})).$$The $\sum^{n}(t_{f},t_{0})$ are the average growth rates over the time interval T between $t_{0}$ and $t_{f}$.

The SVs, unlike the FTNMs and CLVs, depend on the field variables, such as streamfunction, velocity or vorticity, defining the phase-space or norm, for which they are calculated. Transformation between two such field variables denoted x and y can be achieved through
(51)y=Γx
where Γ is the transformation matrix. The general inner product and norm are defined by
(52)$$[x,x^{\prime }]=(y,y^{\prime })=x^{†}\Gamma^{†}\Gamma x^{\prime };\;[x,x]={\left\|y\right\|}^{2}=x^{†}\Gamma^{†}\Gamma x.$$Here, (y,y) is the Euclidean inner product and ‖y‖ the $L_{2}$ norm [80], (Equations (3.6) and (3.7)).

## 6. Lyapunov Vectors and Exponents and the Oseledec Theorem

Lyapunov exponents, which we denote by Ln for n=1,2,…N, characterize the long-time amplification and decay of linear perturbations to dynamical systems and are fundamental properties of these systems. The growth rate, Ln, depends on whether the perturbation is in, or can be constrained to, a subspace of ℝN [72,86,87]. Associated with each Lyapunov exponent, Ln, there is a left or backward OLV, 𝓾n(t)=lim(τ−→−∞)un(t;[t−τ−]), and a right or forward OLV, 𝓿n(t)=lim(τ+→∞)vn(t;[τ+−t]), that are the asymptotic limits of respective SVs, as detailed in Appendix B. In the long-time limits, the singular value exponents, ∑n, converge to Lyapunov exponents Ln.

From the norm dependent OLVs it is possible to construct nonorthogonal but norm independent Lyapunov vectors, $\psi^{n}(t)$, that are propagated forward or backward in time by multiplying by the propagator or its inverse respectively [56,57,58,59,60]. These vectors, $\psi^{n}(t)$, that covary with the dynamics are commonly known as covariant Lyapunov vectors (CLVs) in the terminology of Ginelli et al. [57]. The CLVs, $\psi^{n}(t)$, also grow and contract on average in the forward and backward time directions with the global Lyapunov growth rates $L^{n}$. These important aspects of the behaviour of linear disturbances to dynamical systems are governed by the Oseledec [72] multiplicative ergodic theorem outlined in Appendix B.

## 7. Relationships between Covariant Lyapunov Vectors and Orthogonal Lyapunov Vectors

The relationships between long-time SVs, OLVs and CLVs are summarized in Appendix B based on subspaces of ℝN. Both the case of nondegenerate Lyapunov vectors, with distinct Lyapunov exponents, L1>L2>…>LN, and the degenerate case, where some exponents are equal, L1≥L2≥…≥LN, and there are just M<N distinct Lyapunov exponents, L1>L2>…>LM, are considered.

In this Section, we consider the construction of CLVs from OLVs as well as the construction of OLVs from CLVs. We focus in the next two Subsections on the nondegenerate case, which has mainly been considered in the literature, and in Section 7.3 we discuss the degenerate case. The relationships between the left and right orthonormal Lyapunov vectors (OLVs) and CLVs have been discussed by many authors [56,57,58,59,60,61,62,63,64,65]. A primary aim of much of this work has been the construction of CLVs from long-time SVs that approximate OLVs.

### 7.1. Construction of Orthonormal Lyapunov Vectors from Covariant Lyapunov Vectors

Here we summarize some of the underpinning mathematical relationships between CLVs and OLVs starting with the construction of OLVs from CLVs. We consider the case of nondegenerate real Lyapunov exponents so that the OLVs as well as CLVs are nondegenerate. The left OLVs, ${𝓾}^{n}(t)=\lim(\tau_{-}\to -\infty )u^{n}(t;[t-\tau_{-}])$, may then be obtained by Gram-Smidt orthonormalization of the CLVs, $\psi^{n}(t)$, through the following relationship:(53)$${𝓾}^{n}(t)=\begin{array}{c} \lim\\ \tau_{-}\to -\infty \end{array}u^{n}(t;[t-\tau_{-}])=\frac{\psi^{n}(t)-\sum_{j=1}^{n-1}\left({𝓾}^{j}(t),\psi^{n}(t)){𝓾}^{j}(t\right)}{\left\|\psi^{n}(t)-\sum_{j=1}^{n-1}\left({𝓾}^{j}(t),\psi^{n}(t)){𝓾}^{j}(t\right)\right\|}\;$$
for n=1,2,…,N and where (⋅,⋅) is the Euclidean inner product defined by Equation (52) with ‖⋅‖ the associated $L_{2}$ norm. Similarly, the right OLVs, ${𝓿}^{n}(t)=\lim(\tau_{+}\to \infty )v^{n}(t;[\tau_{+}-t])$, may be determined by orthonormalization of the CLVs, $\psi^{n}(t)$, through
(54)$${𝓿}^{n}(t)=\begin{array}{c} \lim\\ \tau_{+}\to \infty \end{array}v^{n}(t;[\tau_{+}-t])=\frac{\psi^{n}(t)-\sum_{j=n+1}^{N}\left({𝓿}^{j}(t),\psi^{n}(t)){𝓿}^{j}(t\right)}{\left\|\psi^{n}(t)-\sum_{j=n+1}^{N}\left({𝓿}^{j}(t),\psi^{n}(t)){𝓿}^{j}(t\right)\right\|}.$$We note from these relationships that the first CLV and first left OLV are equivalent and the last CLV and last right OLV are equivalent. Given that all the CLVs are norm independent then so are the first left OLV and the last right OLV but the other OLVs are norm dependent. We also see that ${𝓾}^{n}(t)$ only depends on the first j=1,…,n CLVs, $\psi^{j}(t)$, while ${𝓿}^{n}(t)$ only depends on the last j=n,…,N CLVs. These relationships can be written in matrix form:(55)$$U=\Psi B^{U};V=\Psi B^{V}$$
where
(56)$$\Psi =(\psi^{1},\psi^{2},\ldots ,\psi^{N})$$
is the matrix consisting of column vectors $\psi^{n}(t)$ for n=1,2,…,N, and U and V are corresponding matrices of ${𝓾}^{n}(t)$ and ${𝓿}^{n}(t)$. We note that the transformation matrices $B^{U}$ and $B^{V}$ are respectively upper triangular and lower triangular with non-zero elements on the diagonal. They are thus nonsingular with corresponding inverses $C^{U}={\left(B^{U}\right)}^{-1}$ and $C^{V}={\left(B^{V}\right)}^{-1}$ that are respectively upper triangular and lower triangular. That is, inverting the two relationships in Equation (55) we have
(57)$$\Psi =UC^{U}\equiv U{\left(B^{U}\right)}^{-1};\Psi =VC^{V}\equiv V{\left(B^{V}\right)}^{-1}.$$

### 7.2. Construction of Covariant Lyapunov Vectors from Orthonormal Lyapunov Vectors

There are of course efficient methods for constructing the OLVs directly [30,41,63,86,87] and the more difficult task is generally to construct the CLVs efficiently [56,57,58,59,60]. The matrix relationships in Equation (57) can be written in terms of the Lyapunov vectors as follows:(58)ψn(t)=ψ𝓾n(t)≡∑j=1n(𝓾j(t),ψn(t))𝓾j(t)
and
(59)ψn(t)=ψ𝓿n(t)≡∑j=nN(𝓿j(t),ψn(t))𝓿j(t).These relationships state that, as expected, $\psi^{n}(t)$ only depends on the first j=1,…,n left OLVs ${𝓾}^{n}(t)$ and only depends on the last j=n,…,N right OLVs ${𝓿}^{n}(t)$. The restricted ranges of the sums of course also reflect the triangular nature of the transformation matrices and the subspace relationships in Section B.3.

The above formulae for constructing CLVs from OLVs require N OLVs where N may be very large. A more efficient method was developed by Wolfe and Samelson [56] for determining the leading n CLVs, $\psi^{n}(t)$, from the leading n left (backward) OLVs, ${𝓾}^{n}(t)$, and n−1 right (forward) OLVs, ${𝓿}^{n}(t)$. Another efficient algorithm was proposed independently by Ginelli et al. [57] and there have been subsequent variations and improvements by several authors [58,59,60].

### 7.3. Degenerate Orthonormal Lyapunov Vectors

The above results on the relationships between SVs over sufficiently long-time intervals, or OLVs, and CLVs, may be established most directly when the OLVs are nondegenerate and the Oseledec subspaces (Section B.3) are one-dimensional. As noted by Kuptsov and Parlitz [58], if the CLVs have been determined, then the left and right OLVs can still be calculated through Equations (53) and (54). If, on the other hand, we wish to determine the CLVs from the OLVs then Equations (58) and (59) do not determine the CLVs uniquely. This is because the CLVs can have arbitrary orientation in the subspaces associated with identical Lyapunov exponents $L^{n}$. However, the subspaces can be defined by any linearly independent set of vectors and the set determined by Equations (58) and (59) will suffice, as will be shown for dynamics in FTNM-space.

In the subsequent Sections, we shall primarily use Equations (53) to (59) in FTNM-space where they simplify at critical times. As noted in Section 3 and Appendix A, we consider nonsingular propagators (as in Oseledec [72]) for which the FTNMs, $\phi^{n}(t)$, are nondegenerate with distinct but possibly complex eigenvalues and exponents $\Lambda^{n}=\Lambda_{r}^{n}+i\Lambda_{i}^{n}$. This means that not all the real parts of the eigenvalue exponents $\Lambda_{r}^{n}$ are distinct. As consequence, in FTNM-space, some of the singular value exponents, (Equation (76)), ${\underline{\sum }}^{n}=\Lambda_{r}^{n}$, are not distinct and the SVs are degenerate. As noted above and in Section 8, this is not a problem since we can always choose the SVs to be ordered in the same way as the FTNMs. Furthermore, this also applies in the long-time limits when the SVs approach the OLVs and some of the Lyapunov exponents $L^{n}$ are identical ( with M=M and 𝒹(m)=d(m)), as detailed in the following Sections.

## 8. Dynamics in FTNM-Space

The evolution of SVs in physical space can be highly complex and very different depending on the norm or field variables for which they are defined [50]. Much of this complicated behaviour can be understood in terms of the projection of SVs in terms of the norm independent FTNMs [50]. In this Section, we consider the relationships between SVs and FTNMs when the dynamical equations are transformed to the space spanned by FTNMs.

### Tangent Linear Equation and Propagator in FTNM-Space

We underline the variables, vectors, and matrices in FTNM-space to distinguish them from the corresponding quantities in the original x-space:(60)$$x(t)=\Phi (t_{0};[t_{f}-t_{0}])\underline{x}(t)$$
and
(61)$$\Phi (t_{0};[t_{f}-t_{0}])=(\phi^{1}(t_{0};[t_{f}-t_{0}]),\phi^{2}(t_{0};[t_{f}-t_{0}]),\ldots ,\phi^{N}(t_{0};[t_{f}-t_{0}]))$$
where $\phi^{n}(t_{0})$ are the FTNMs of Section 3. From Equation (17) we see that the components of $\underline{x}(t_{0})$ are related to the projection coefficients in terms of FTNMs through ${\underline{x}}_{n}(t_{0})=\kappa_{n}$ for n=1,…,N. In FTNM-space the prognostic equation for $\underline{x}(t)$ becomes
(62)$$\frac{d\underline{x}(t)}{dt}=\underline{M}(t)\underline{x}(t)$$
where the dynamical matrix $\underline{M}(t)$ is given by
(63)$$\underline{M}(t)=\Phi^{-1}(t_{0};[t_{f}-t_{0}])M(t)\Phi (t_{0};[t_{f}-t_{0}]).$$As noted in Section 3, we assume that Φ is non-singular so that $\Phi^{-1}$ is well defined. Again
(64)$$\underline{x}(t)=\underline{G}(t,t_{0})\underline{x}(t_{0})$$
where $\underline{G}(t,t_{0})$ is the propagator from the initial time $t_{0}$ to time t in FTNM-space. The propagator in FTNM-space also satisfies Equations (5) to (9), including the central cocycle properties, for the corresponding underlined matrices. Importantly,
(65)$$\underline{G}(t_{f},t_{0})=\Phi^{-1}G(t_{f},t_{0})\Phi =D_{\lambda }(t_{f},t_{0})=\mathrm{diag}(\lambda^{1}(t_{f},t_{0}),\lambda^{2}(t_{f},t_{0}),\ldots ,\lambda^{N}(t_{f},t_{0}))$$
and
(66)$$\begin{array}{l} {\underline{G}}^{†}(t_{f},t_{0})=\Phi^{†}G^{†}(t_{f},t_{0})({\Phi^{-1})}^{†}=A^{-1}G^{†}(t_{f},t_{0})A\\ =D_{\lambda^{*}}(t_{f},t_{0})=\mathrm{diag}(\lambda^{1*}(t_{f},t_{0}),\lambda^{2*}(t_{f},t_{0}),\ldots ,\lambda^{N*}(t_{f},t_{0})). \end{array}$$We also have
(67)$$\begin{array}{l} \underline{G}(t_{f},t_{0}){\underline{G}}^{†}(t_{f},t_{0})={\underline{G}}^{†}(t_{f},t_{0})\underline{G}(t_{f},t_{0})=D_{\lambda^{*}}(t_{f},t_{0})D_{\lambda }(t_{f},t_{0})=D_{{\underline{\sigma }}^{2}}(t_{f},t_{0})\\ =\mathrm{diag}(\lambda^{1}\lambda^{1*},\lambda^{2}\lambda^{2*},\ldots ,\lambda^{N}\lambda^{N*})=\mathrm{diag}(({\underline{\sigma }}^{1}{)}^{2},({{\underline{\sigma }}^{2})}^{2},\ldots ,({{\underline{\sigma }}^{N})}^{2}) \end{array}$$
where $\lambda^{n}\equiv {\underline{\lambda }}^{n}=\lambda^{n}(t_{f},t_{0})$ and ${\underline{\sigma }}^{n}={\underline{\sigma }}^{n}(t_{f},t_{0})=\left|\lambda^{n}(t_{f},t_{0})\right|$.

In FTNM-space Equation (11) again holds for the associated underlined variables:(68)$$\begin{array}{l} \underline{G}(t_{f},t_{0}){\underline{\phi }}^{n}(t_{0};[t_{f}-t_{0}])=D_{\lambda }(t_{f},t_{0}){\underline{\phi }}^{n}(t_{0};[t_{f}-t_{0}])=\lambda^{n}(t_{f},t_{0}){\underline{\phi }}^{n}(t_{0};[t_{f}-t_{0}])\\ ={\underline{\phi }}^{n}(t_{f};[t_{f}-t_{0}]) \end{array}$$
with n=1,2,…,N. Because $\underline{G}(t_{f},t_{0})=D_{\lambda }(t_{f},t_{0})$ is now a normal matrix, in fact a diagonal matrix, the FTNMs can be taken to be proportional to the standard unit basis vectors:(69)$${\underline{\phi }}^{n}(t_{0};[t_{f}-t_{0}])={\underline{\phi }}_{n}^{n}(t_{0})e^{n}\;;\;{\underline{\phi }}^{n}(t_{f};[t_{f}-t_{0}])={\underline{\phi }}_{n}^{n}(t_{f})e^{n}.$$Indeed, the eigenmodes ${\underline{\phi }}^{n}$, adjoint eigenmodes ${\underline{\alpha }}^{n}$, and left ${\underline{u}}^{n}$ and right ${\underline{v}}^{n}$ SVs can all be taken to be proportional to the standard unit basis vectors $e^{n}$. Here the components $e_{m}^{n}$ of $e^{n}$ are the Kronecker delta function
(70)$$e_{m}^{n}=\delta_{mn}=\left\{ \begin{array}{l} 1\;\mathrm{if}\;m=n\\ 0\;\mathrm{otherwise}. \end{array}\right.$$

In FTNM-space Equations (46) and (47) become
(71)$$\begin{array}{l} \underline{G}(t_{f},t_{0}){\underline{G}}^{†}(t_{f},t_{0}){\underline{u}}^{n}(t_{f};[t_{f}-t_{0}])=D_{\lambda }^{}(t_{f},t_{0})D_{\lambda^{*}}^{}(t_{f},t_{0}){\underline{u}}^{n}(t_{f};[t_{f}-t_{0}])\\ =D_{{\underline{\sigma }}^{2}}^{}(t_{f},t_{0}){\underline{u}}^{n}(t_{f};[t_{f}-t_{0}])={\left({\underline{\sigma }}^{n}(t_{f},t_{0})\right)}^{2}{\underline{u}}^{n}(t_{f};[t_{f}-t_{0}]) \end{array}$$
and
(72)$$\begin{array}{l} {\underline{G}}^{†}(t_{f},t_{0})\underline{G}(t_{f},t_{0}){\underline{v}}^{n}(t_{0};[t_{f}-t_{0}])=D_{\lambda^{*}}^{}(t_{f},t_{0})D_{\lambda }^{}(t_{f},t_{0}){\underline{v}}^{n}(t_{0};[t_{f}-t_{0}])\\ =D_{{\underline{\sigma }}^{2}}^{}(t_{f},t_{0}){\underline{v}}^{n}(t_{0};[t_{f}-t_{0}])={\left({\underline{\sigma }}^{n}(t_{f},t_{0})\right)}^{2}{\underline{v}}^{n}(t_{0};[t_{f}-t_{0}]). \end{array}$$Thus, we can take
(73)$${\underline{u}}^{n}(t_{f};[t_{f}-t_{0}])={\tilde{\underline{\phi }}}_{n}^{n}(t_{f};[t_{f}-t_{0}])e^{n}\equiv \frac{{\underline{\phi }}_{n}^{n}(t_{f};[t_{f}-t_{0}])e^{n}}{\left|{\underline{\phi }}_{n}^{n}(t_{f};[t_{f}-t_{0}])\right|}={\underline{u}}_{n}^{n}(t_{f};[t_{f}-t_{0}])e^{n}$$
with $\left|{\tilde{\underline{\phi }}}_{n}^{n}(t_{f};[t_{f}-t_{0}])\right|=1=\left|{\underline{u}}_{n}^{n}(t_{f};[t_{f}-t_{0}])\right|$ for n=1,2,…,N. As well,
(74)$${\underline{v}}^{n}(t_{0};[t_{f}-t_{0}])={\tilde{\underline{\phi }}}_{n}^{n}(t_{0};[t_{f}-t_{0}])e^{n}\equiv \frac{{\underline{\phi }}_{n}^{n}(t_{0};[t_{f}-t_{0}])e^{n}}{\left|{\underline{\phi }}_{n}^{n}(t_{0};[t_{f}-t_{0}])\right|}={\underline{v}}_{n}^{n}(t_{0};[t_{f}-t_{0}])e^{n}\;$$
with $\left|{\tilde{\underline{\phi }}}_{n}^{n}(t_{0};[t_{f}-t_{0}])\right|=1=\left|{\underline{v}}_{n}^{n}(t_{0};[t_{f}-t_{0}])\right|$. Here it is important to note from Equation (67) that
(75)$${\underline{\sigma }}^{n}(t_{f},t_{0})=\left|\lambda^{n}(t_{f},t_{0})\right|$$
and from Equations (14) and (50)
(76)$${\underline{\sum }}^{n}(t_{f},t_{0})=\Lambda_{r}^{n}(t_{f},t_{0}).$$

As noted in Section 2, we assume that the in general complex eigenvalues, $\lambda^{n}$, are unique while the real singular values ${\underline{\sigma }}^{n}$ may be degenerate, as they would be, for example, when $\lambda^{n+1}=\lambda^{n*}$. Note that the possible degeneracy of the SVs does not affect the ability to choose the basis vectors as in Equations (73) and (74). Indeed, irrespective of the degeneracy of the SVs and multiplicity of the singular values, and corresponding exponents, we can always choose the ordering to be the same as for the FTNMs and their eigenvalues and exponents detailed in Appendix A. This also applies when the time spans considered are sufficiently long for the SVs to approach the orthonormal Lyapunov vectors as seen from Appendix A and Section B.4 (with M=M and 𝒹(m)=d(m)).

## 9. Covariant Lyapunov Vectors for Periodic Systems

In this section we explore the relationship between OLVs and CLVs for periodic systems in FTNM-space before studying aperiodic systems in the next Section. The simplifications that occur in FTNM-space at critical times straightforwardly allows the direct determination of these relationships, from Equations (53) to (59), and the equivalence of Floquet vectors [55] and CLVs [65,88]. Because of the anchoring of SVs with FTNMs at critical times, these links are simply established for degenerate as well as nondegenerate Lyapunov spectra. We consider the case when the FTNMs are defined on the basic period of the Floquet vectors.

### 9.1. Dynamics of SVs and OLVs in FTNM-Space

We start by considering the relationships between SVs and FTNMs in FTNM-space for periodic systems where the analysis is simplified by using the results established in Section 8. Over the basic period $T=T(t_{f},t_{0})=[t_{f}-t_{0}]$ in Equation (10), FTNMs are eigenvectors of the propagator as in Equation (68) and the FTNMs are proportional to the standard unit basis vectors (Equation (70)) through the relationships in Equation (69). The same is true for periods that are multiples of the basic period. We define
(77)$$\tau_{\pm }(\hat{t})=\hat{t}\pm KT\;\mathrm{for}\;\hat{t}=t_{0}\;\mathrm{or}\;\hat{t}=t_{f}=t_{0}+T$$
where K is a positive integer. Consider the propagator over longer time intervals:(78)$$\begin{array}{l} \underline{G}(\hat{t},\tau_{-}(\hat{t}))\to {\left[\underline{G}(t_{f},t_{0})\right]}^{K}=D_{\lambda }^{K}(t_{f},t_{0}),\\ {\underline{G}}^{†}(\tau_{+}(\hat{t}),\hat{t})\to {\left[{\underline{G}}^{†}(t_{f},t_{0})\right]}^{K}=D_{\lambda^{*}}^{K}(t_{f},t_{0}). \end{array}$$Then
(79)$$\begin{array}{l} \underline{G}(\hat{t},\tau_{-}(\hat{t})){\underline{\phi }}^{n}(\tau_{-}(\hat{t});[\hat{t}-\tau_{-}(\hat{t})])=D_{\lambda }^{K}(t_{f},t_{0}){\underline{\phi }}^{n}(\tau_{-}(\hat{t});[\hat{t}-\tau_{-}(\hat{t})])\\ ={\left[\lambda^{n}(t_{f},t_{0})\right]}^{K}{\underline{\phi }}^{n}(\tau_{-}(\hat{t});[\hat{t}-\tau_{-}(\hat{t})])={\underline{\phi }}^{n}(\hat{t};[\hat{t}-\tau_{-}(\hat{t})]) \end{array}$$
and
(80)$$\begin{array}{l} \underline{G}(\tau_{+}(\hat{t}),\hat{t}){\underline{\phi }}^{n}(\hat{t};[\tau_{+}(\hat{t})-\hat{t}])=D_{\lambda }^{K}(t_{f},t_{0}){\underline{\phi }}^{n}(\hat{t};[\tau_{+}(\hat{t})-\hat{t}])\\ ={\left[\lambda^{n}(t_{f},t_{0})\right]}^{K}{\underline{\phi }}^{n}(\hat{t};[\tau_{+}(\hat{t})-\hat{t}])={\underline{\phi }}^{n}(\tau_{+}(\hat{t});[\tau_{+}(\hat{t})-\hat{t}]). \end{array}$$Again, the initial and final FTNMs are proportional to the standard unit basis vectors and, in particular
(81)$${\underline{\phi }}^{n}(\hat{t};[\hat{t}-\tau_{-}(\hat{t})])={\underline{\phi }}_{n}^{n}(\hat{t})e^{n}\;;\;{\underline{\phi }}^{n}(\hat{t};[\tau_{+}(\hat{t})-\hat{t}])={\underline{\phi }}_{n}^{n}(\hat{t})e^{n}.$$

We consider next the SVs over the longer time intervals. We see from Equations (46) and (47) but for the more general times in Equation (77) that
(82)$$\begin{array}{l} \underline{G}(\hat{t},\tau_{-}(\hat{t})){\underline{G}}^{†}(\hat{t},\tau_{-}(\hat{t})){\underline{u}}^{n}(\hat{t};[\hat{t}-\tau_{-}(\hat{t})])=D_{{\underline{\sigma }}^{2}}^{K}(t_{f},t_{0}){\underline{u}}^{n}(\hat{t};[\hat{t}-\tau_{-}(\hat{t})])\\ ={\left[{\underline{\sigma }}^{n}(t_{f},t_{0})\right]}^{2K}{\underline{u}}^{n}(\hat{t};[\hat{t}-\tau_{-}(\hat{t})])=({\underline{\sigma }}^{n}{\left(\hat{t},\tau_{-}(\hat{t})\right)}^{2}{\underline{u}}^{n}(\hat{t};[\hat{t}-\tau_{-}(\hat{t})]) \end{array}$$
and
(83)$$\begin{array}{l} {\underline{G}}^{†}(\tau_{+}(\hat{t}),\hat{t})\underline{G}(\tau_{+}(\hat{t}),\hat{t}){\underline{v}}^{n}(\hat{t};[\tau_{+}(\hat{t})-\hat{t}])=D_{{\underline{\sigma }}^{2}}^{K}(t_{f},t_{0}){\underline{v}}^{n}(\hat{t};[\tau_{+}(\hat{t})-\hat{t}])\\ ={\left[{\underline{\sigma }}^{n}(t_{f},t_{0})\right]}^{2K}{\underline{v}}^{n}(\hat{t};[\tau_{+}(\hat{t})-\hat{t}])={\left({\underline{\sigma }}^{n}(\tau_{+}(\hat{t}),\hat{t})\right)}^{2}{\underline{v}}^{n}(\hat{t};[\tau_{+}(\hat{t})-\hat{t}]) \end{array}$$
with
(84)$$D_{\lambda^{*}}^{K}(t_{f},t_{0})D_{\lambda }^{K}(t_{f},t_{0})=D_{{\underline{\sigma }}^{2}}^{K}(t_{f},t_{0}).$$As noted in Section 8, Equations (73) and (74), because $\underline{G}=D_{\lambda }$ over any multiple of the basic period $T=T(t_{f},t_{0})=[t_{f}-t_{0}]$ is a normal matrix, in fact diagonal, the left and right SVs are also proportional to standard unit basis vectors $e^{n}$ of Equation (70). Thus, for $\hat{t}=t_{0}\;\mathrm{or}\;\hat{t}=t_{f}$,
(85)$${\underline{u}}^{n}(\hat{t};[\hat{t}-\tau_{-}(\hat{t})])=\frac{{\underline{\phi }}_{n}^{n}(\hat{t};[\hat{t}-\tau_{-}(\hat{t})])e^{n}}{\left|{\underline{\phi }}_{n}^{n}(\hat{t};[\hat{t}-\tau_{-}(\hat{t})])\right|}\equiv {\tilde{\underline{\phi }}}_{n}^{n}(\hat{t};[\hat{t}-\tau_{-}(\hat{t})])e^{n}={\underline{u}}_{n}^{n}(\hat{t};[\hat{t}-\tau_{-}(\hat{t})])e^{n},$$
with unit norm FTNMs (and CLV vectors) and components denoted by $\tilde{}$ so that $\left|{\tilde{\underline{\phi }}}_{n}^{n}(\hat{t};[\hat{t}-\tau_{-}(\hat{t})])\right|=1=\left|{\underline{u}}_{n}^{n}(\hat{t};[\hat{t}-\tau_{-}(\hat{t})])\right|$. Similarly,
(86)$${\underline{v}}^{n}(\hat{t};[\tau_{+}(\hat{t})-\hat{t}])=\frac{{\underline{\phi }}_{n}^{n}(\hat{t};[\tau_{+}(\hat{t})-\hat{t}])e^{n}}{\left|{\underline{\phi }}_{n}^{n}(\hat{t};[\tau_{+}(\hat{t})-\hat{t}])\right|}\equiv {\tilde{\underline{\phi }}}_{n}^{n}(\hat{t};[\tau_{+}(\hat{t})-\hat{t}])e^{n}={\underline{v}}_{n}^{n}(\hat{t};[\tau_{+}(\hat{t})-\hat{t}])e^{n}\;$$
where $\;\left|{\underline{\tilde{\phi }}}_{n}^{n}(\hat{t};[\tau_{+}(\hat{t})-\hat{t}])\right|=1=\left|{\underline{v}}_{n}^{n}(\hat{t};[\tau_{+}(\hat{t})-\hat{t}])\right|$. Also, from Equations (39), (75), (82) and (83) it follows that
(87)$${\underline{\sigma }}^{n}(t_{f},\tau_{-}(t_{f}))={\underline{\sigma }}^{n}(t_{0},\tau_{-}(t_{0}))={\underline{\sigma }}^{n}(\tau_{+}(t_{0}),t_{0})={\underline{\sigma }}^{n}(\tau_{+}(t_{f}),t_{f})={\left({\underline{\sigma }}^{n}(t_{f},t_{0})\right)}^{K}={\left|\lambda^{n}(t_{f},t_{0})\right|}^{K}.$$Now, with $t^{\prime }$ denoting any of the above initial times and t any of the above final times we have
(88)$$\lambda^{n}(t,t^{\prime })=\lambda_{r}^{n}+i\lambda_{i}^{n}=\exp\Lambda^{n}(t,t^{\prime })(t-t^{\prime })=\exp\{\Lambda_{r}^{n}(t,t^{\prime })+i\Lambda_{i}^{n}(t,t^{\prime })\}(t-t^{\prime })$$
and
(89)$${\underline{\sigma }}^{n}(t,t^{\prime })=\exp{\underline{\sum }}^{n}(t,t^{\prime })(t-t^{\prime }).$$Of course, the above relationships also apply in the limit where K→∞, so that the left OLV
(90)$${\underline{𝓾}}^{n}(\hat{t})={\underline{u}}^{n}(\hat{t};-\infty )=\frac{{\underline{\phi }}_{}^{n}(\hat{t})}{\left\|{\underline{\phi }}_{}^{n}(\hat{t})\right\|}=\frac{{\underline{\phi }}_{n}^{n}(\hat{t})e^{n}}{\left|{\underline{\phi }}_{n}^{n}(\hat{t})\right|}\equiv {\tilde{\underline{\phi }}}_{n}^{n}(\hat{t})e^{n}={\underline{𝓊}}_{n}^{n}(\hat{t})e^{n}$$
with $\left|{\tilde{\underline{\phi }}}_{n}^{n}(\hat{t})\right|=1=\left|{\underline{𝓊}}_{n}^{n}(\hat{t})\right|$ for n=1,2,…,N and $\hat{t}=t_{0}\;\mathrm{or}\;\hat{t}=t_{f}$. Also, the right OLV
(91)$${\underline{𝓿}}^{n}(\hat{t})={\underline{v}}^{n}(\hat{t};\infty )=\frac{{\underline{\phi }}_{}^{n}(\hat{t})}{\left\|{\underline{\phi }}_{}^{n}(\hat{t})\right\|}=\frac{{\underline{\phi }}_{n}^{n}(\hat{t})e^{n}}{\left|{\underline{\phi }}_{n}^{n}(\hat{t})\right|}\equiv {\tilde{\underline{\phi }}}_{n}^{n}(\hat{t})e^{n}={\underline{𝓋}}_{n}^{n}(\hat{t})e^{n}$$
with $\left|{\tilde{\underline{\phi }}}_{n}^{n}(\hat{t})\right|=1=\left|{\underline{𝓋}}_{n}^{n}(\hat{t})\right|$. As well the Lyapunov exponent
(92)$$L^{n}=\Lambda_{r}^{n}(t_{f},t_{0})={\underline{\sum }}^{n}(t_{f},t_{0})={\underline{\sum }}^{n}(t_{f},-\infty )={\underline{\sum }}^{n}(t_{0},-\infty )={\underline{\sum }}^{n}(\infty ,t_{0})={\underline{\sum }}^{n}(\infty ,t_{f}).$$Here, we note that ${\underline{\phi }}^{n}(\hat{t})={\underline{\phi }}^{n}(\hat{t};\infty )={\underline{\phi }}^{n}(\hat{t};-\infty )={\underline{\phi }}^{n}(\hat{t};[t_{f}-t_{0}])$ for n=1,2,…,N and $\hat{t}=t_{0}\;\mathrm{or}\;\hat{t}=t_{f}$. In the above analysis we have not used Equations (53) and (54) but just relied on the long-time behaviour of SVs in FTNM-space.

### 9.2. Construction of CLVs from OLVs in FTNM-Space

Next, we use the results in Section 7.2 to construct the CLVs from the OLVs in FTNM-space. From Equations (53) and (90) we have
(93)ψ_n(t^)=ψ_𝓾n(t^)=∑j=1n(𝓊_jj(t^)ej,ψ_n(t^))𝓊_jj(t^)ej=∑j=1nψ_jn(t^)ej
and from Equations (54) and (91)
(94)ψ_n(t^)=ψ_𝓿n(t^)=∑j=nN(𝓋_jj(t^)ej,ψ_n(t^))𝓋_jj(t^)ej=∑j=nNψ_jn(t^)ej.These results then mean that the only nonvanishing elements are ${\underline{\psi }}_{n}^{n}(\hat{t})$ and that
(95)ψ_n(t^)=ψ_𝓾n(t^)=(𝓾_n(t^),ψ_n(t^))𝓾_n(t^)=(ϕ_˜nn(t^)en,ψ_nn(t^)en)ϕ_˜nn(t^)en=ψ_nn(t^)en
with unit norm FTNM and CLV vectors and components denoted by $\tilde{}$ so that ${\tilde{\underline{\phi }}}_{n}^{n}(\hat{t})={\underline{\phi }}_{n}^{n}(\hat{t})/\left|{\underline{\phi }}_{n}^{n}(\hat{t})\right|$ and
(96)ψ_n(t^)=ψ_𝓿n(t^)=(𝓿_n(t^),ψ_n(t^))𝓿_n(t^)=(ϕ_˜nn(t^)en,ψ_nn(t^)en)ϕ_˜nn(t^)en=ψ_nn(t^)en.Moreover, from Equations (95) and (96) we can choose the orientations and amplitudes so that
(97)$${\underline{\psi }}^{n}(\hat{t})={\underline{\psi }}_{n}^{n}(\hat{t})e^{n}={\underline{\phi }}_{n}^{n}(\hat{t})e^{n}={\underline{\phi }}^{n}(\hat{t})$$
for $\hat{t}=t_{0}\;$ or $\;\hat{t}=t_{f}$ and n=1,2,…,N.

Of course, both the CLVs and FTNMs covary with the tangent linear dynamics so that they can be obtained at any future time t or past time $t^{\prime }$ through
(98)$${\underline{\psi }}^{n}(t)=\underline{G}(t,\hat{t}){\underline{\psi }}^{n}(\hat{t})={\underline{\phi }}^{n}(t)=\underline{G}(t,\hat{t}){\underline{\phi }}^{n}(\hat{t})$$
and
(99)$${\underline{\psi }}^{n}(t^{\prime })={\left[\underline{G}(\hat{t},t^{\prime })\right]}^{-1}{\underline{\psi }}^{n}(\hat{t})={\underline{\phi }}^{n}(t^{\prime })={\left[\underline{G}(\hat{t},t^{\prime })\right]}^{-1}{\underline{\phi }}^{n}(\hat{t}).$$The real Lyapunov exponents are independent of finite time t or $t^{\prime }$ and, as in Equation (92), are given by $L^{n}={\underline{\sum }}^{n}(t_{f},t_{0})=\Lambda_{r}^{n}(t_{f},t_{0})$. The CLV exponents are in general complex and equal to the FTNM exponents $\Lambda^{n}(t_{f},t_{0})=\Lambda_{r}^{n}(t_{f},t_{0})+i\Lambda_{i}^{n}(t_{f},t_{0})$ for n=1,2,…,N. These results are valid for both degenerate and nondegenerate Lyapunov spectra as follows simply from the anchoring of the SVs with FTNMs at critical times in FTNM-space. In the original x-space $\psi^{n}(t)=\phi^{n}(t)$, as expected and detailed in Appendix C and Appendix D (see also Section 10.2).

### 9.3. Construction of OLVs from CLVs in FTNM-Space

As noted in the Introduction and in Section 7.2, there are efficient methods for determining OLVs directly and for the periodic problem this can be done very simply in FTNM-space, at critical times, as shown in Section 9.1. Here, we also show that the general results between CLVs and OLVs in Section 7 simplify in FTNM-space to construct OLVs from CLVs.

In FTNM-space Equation (53) reduces to
(100)$${\underline{𝓾}}^{n}(\hat{t})=\frac{{\underline{\psi }}^{n}(\hat{t})}{\left\|{\underline{\psi }}^{n}(\hat{t})\right\|}=\frac{{\underline{\psi }}_{n}^{n}(\hat{t})e^{n}}{\left|{\underline{\psi }}_{n}^{n}(\hat{t})\right|}\equiv {\tilde{\underline{\psi }}}_{n}^{n}(\hat{t})e^{n}=\frac{{\underline{\phi }}_{n}^{n}(\hat{t})e^{n}}{\left|{\underline{\phi }}_{n}^{n}(\hat{t})\right|}\equiv {\tilde{\underline{\phi }}}_{n}^{n}(\hat{t})e^{n}={\underline{𝓊}}_{n}^{n}(\hat{t})e^{n}$$
with $\left|{\tilde{\underline{\phi }}}_{n}^{n}(\hat{t})\right|=1=\left|{\tilde{\underline{\psi }}}_{n}^{n}(\hat{t})\right|=\left|{\underline{𝓊}}_{n}^{n}(\hat{t})\right|$ and for $\hat{t}=t_{0}\;$ or $\;\hat{t}=t_{f}=t_{0}+T$ and n=1,2,…,N where we have used the relationships in Equation (97). As well, for these values of $\hat{t}$ and n, Equation (54) simplifies to
(101)$${\underline{𝓿}}^{n}(\hat{t})=\frac{{\underline{\psi }}^{n}(\hat{t})}{\left\|{\underline{\psi }}^{n}(\hat{t})\right\|}=\frac{{\underline{\psi }}_{n}^{n}(\hat{t})e^{n}}{\left|{\underline{\psi }}_{n}^{n}(\hat{t})\right|}\equiv {\tilde{\underline{\psi }}}_{n}^{n}(\hat{t})e^{n}=\frac{{\underline{\phi }}_{n}^{n}(\hat{t})e^{n}}{\left|{\underline{\phi }}_{n}^{n}(\hat{t})\right|}\equiv {\tilde{\underline{\phi }}}_{n}^{n}(\hat{t})e^{n}={\underline{𝓋}}_{n}^{n}(\hat{t})e^{n}$$
with $\left|{\tilde{\underline{\phi }}}_{n}^{n}(\hat{t})\right|=1=\left|{\tilde{\underline{\psi }}}_{n}^{n}(\hat{t})\right|=\left|{\underline{𝓋}}_{n}^{n}(\hat{t})\right|$. These results of course agree with those established directly in Section 9.1.

The above analysis establishes the left and right OLVs based on the CLVs with Equation (53) reducing to Equation (100) and Equation (54) to Equation (101). We shall not need the expressions for OLVs at other times than those considered here and in Section 9.1 since our primary interest is the construction of CLVs.

## 10. Covariant Lyapunov Vectors for Aperiodic Systems

Next, we consider the determination of Lyapunov exponents and vectors for the case of aperiodic flows under the conditions, described in Section 2 and Appendix B and Appendix C, when the Oseledec [72] theorem holds. We consider both the theoretical principles and the practical numerical calculation of these instability properties. The numerical calculation of Lyapunov exponents and vectors in practice involves the determination of approximations to these dynamical quantities over a finite time interval and with finite accuracy of the propagators. However, we assume the propagators to be accurate for the purpose of establishing theoretical results. Our primary interest is again the determination of CLVs that we want to establish in a time interval $\tau_{+}>\Delta_{+}\geq t\geq \Delta_{-}>\tau_{-}$. For numerical studies this time interval is determined through experimentation. For our theoretical results it is sufficient to take $\Delta_{\pm }=\gamma_{\pm }\tau_{\pm }$ with $1>\gamma_{\pm }>0$ for $\tau_{\pm }\to \pm \infty$ although practical convergence is likely faster with $1\gg \gamma_{\pm }>0$.

We suppose that we have established, to within the errors that we can tolerate, that in the space of field variables and for the norm of interest, the long-time left, and right SVs are suitable approximations to the left and right OLVs. As well, the associated singular value exponents should have closely approached the Lyapunov exponents. Thus, for n=1,2,…,N we have
(102)$${𝓾}^{n}(t)≐u^{n}(t;[t-\tau_{-}])$$
for $\tau_{+}\geq t\geq \Delta_{-}$ and
(103)$${𝓿}^{n}(t)≐v^{n}(t;[\tau_{+}-t])$$
for $\Delta_{+}\geq t\geq \tau_{-}$. Here, we suppose that $\tau_{\pm }=\pm \left|\tau_{\pm }\right|$ and we expect convergence for large $\left|\tau_{\pm }\right|$.

### 10.1. Dynamics in FTNM-Space

We suppose that the convergence of the SVs to the OLVs applies for dynamics in FTNM-space with time interval $\left[\tau_{+}-\tau_{-}\right]$ and, to be specific, for the Euclidean inner product and $L_{2}$ norm. Again, from the development in Section 8, and particularly Equations (65) to (70), (with $t_{0}\to \tau_{-}$ and $t_{f}\to \tau_{+}$) so that $\Phi =\Phi (\tau_{-};[\tau_{+}-\tau_{-}])$ we see that
(104)$${\underline{\phi }}^{n}(\tau_{-};[\tau_{+}-\tau_{-}])={\underline{\phi }}_{n}^{n}(\tau_{-})e^{n},$$
and
(105)$${\underline{\phi }}^{n}(\tau_{+};[\tau_{+}-\tau_{-}])={\underline{\phi }}_{n}^{n}(\tau_{+})e^{n}$$
where the FTNMs at the end points are proportional to the standard unit basis vectors. Of course, if $\left[\tau_{+}-\tau_{-}\right]$ should correspond to a sufficiently long period, or multiple periods, of a periodic orbit very close to the aperiodic trajectory then we would be back to the situation in Section 9.

As noted by Poincare [89] and recounted by many subsequent researchers [65,90,91] there are always periodic orbits that are very close to aperiodic trajectories. Van Veen et al. [91] translate Poincare’s insight as: “Given the equations … and a particular solution one can always find a periodic solution (of which the period may indeed be very long) such that the difference between the two remains as small as one likes for as long as one likes”. Our aim here has just been to note that there are situations for aperiodic flows where one might expect to be able to approximate CLVs by FTNMs and where the relationships between OLVs and FTNMs simplify (at critical times) in FTNM-space.

Next, we approach the problem of determining approximations to CLVs from long time left and right SVs in FTNM-space that in turn are estimates of the OLVs, Thus, from Equations (73) and (102) the left or backward OLVs are determined by
(106)$${\underline{𝓾}}^{n}(\tau_{+})≐{\underline{u}}^{n}(\tau_{+};[\tau_{+}-\tau_{-}])=\frac{{\underline{\phi }}_{n}^{n}(\tau_{+};[\tau_{+}-\tau_{-}])e^{n}}{\left|{\underline{\phi }}_{n}^{n}(\tau_{+};[\tau_{+}-\tau_{-}])\right|}\equiv {\tilde{\underline{\phi }}}_{n}^{n}(\tau_{+};[\tau_{+}-\tau_{-}])e^{n}≐{\underline{𝓊}}_{n}^{n}(\tau_{+})e^{n}\;$$
with $\left|{\tilde{\underline{\phi }}}_{n}^{n}(\tau_{+})\right|=1=\left|{\underline{𝓊}}_{n}^{n}(\tau_{+})\right|\;$for n=1,2,…,N. Similarly, from Equations (74) and (103), the right or forward OLVs are given by
(107)$${\underline{𝓿}}^{n}(\tau_{-})≐{\underline{v}}^{n}(\tau_{-};[\tau_{+}-\tau_{-}])=\frac{{\underline{\phi }}_{n}^{n}(\tau_{-};[\tau_{+}-\tau_{-}])e^{n}}{\left|{\underline{\phi }}_{n}^{n}(\tau_{-};[\tau_{+}-\tau_{-}])\right|}\equiv {\tilde{\underline{\phi }}}_{n}^{n}(\tau_{-};[\tau_{+}-\tau_{-}])e^{n}≐{\underline{𝓋}}_{n}^{n}(\tau_{-})e^{n}\;$$
with $\left|{\tilde{\underline{\phi }}}_{n}^{n}(\tau_{-})\right|=1=\left|{\underline{𝓋}}_{n}^{n}(\tau_{-})\right|$. Now, Equation (58) for the CLVs simplifies to
(108)ψ_n(τ+)=ψ_𝓾n(τ+)≡∑j=1n(𝓾_j(τ+),ψ_n(τ+))𝓾_j(τ+)≐∑j=1n(ϕ_˜jj(τ+)ej,ψ_n(τ+))ϕ_˜jj(τ+)ej≐∑j=1nψ_jn(τ+)ej
and Equation (59) reduces to
(109)ψ_n(τ−)=ψ_𝓿n(τ−)≡∑j=nN(𝓿_j(τ−),ψn(τ−))𝓿_j(τ−)≐∑j=nN(ϕ_˜jj(τ−)ej,ψ_n(τ−))ϕ_˜jj(τ−)ej≐∑j=nNψ_jn(τ−)ej.

More generally, with both right and left OLVs approximated sufficiently accurately by the corresponding SVs, as in Equations (102) and (103), for $\Delta_{+}\geq t\geq \Delta_{-}$, we can determine estimations for the CLVs from Equations (58) and (59) in this time interval. Equivalently, because of the covariant properties of ${\underline{\psi }}^{n}(t)$, the two expressions in Equations (108) and (109) can be propagated into the time interval $\Delta_{+}\geq t\geq \Delta_{-}$ and related as
(110) ψ_n(t)≐G_(t,τ−)ψ_𝓿n(τ−)≐[G_(τ+,t)]−1ψ_𝓾n(τ+)≐G_(t,τ−)∑j=nNψ_jn(τ−)ej≐G_(t,τ−)[Dλ]−1∑j=1nψ_jn(τ+)ej=G_(t,τ−)∑j=1n(λj)−1ψ_jn(τ+)ej.Here, $\lambda^{j}=\lambda^{j}(\tau_{+},\tau_{-})$, and we have used the fact that, as noted in Section 2, the propagator is assumed to be nonsingular so that both $\underline{G}(t,t^{\prime })$ and ${\left[\underline{G}(t,t^{\prime })\right]}^{-1}$ are well defined [72] for $\tau_{+}\geq t\geq t^{\prime }\geq \tau_{-}$. Now, the coefficients multiplying $e^{j}$ in Equation (110) must equate and thus the only non-zero terms are for j=n: (111)$${\underline{\psi }}^{n}(t)≐\underline{G}(t,\tau_{-}){\underline{\psi }}_{n}^{n}(\tau_{-})e^{n}≐{\left[\underline{G}(\tau_{+},t)\right]}^{-1}{\underline{\psi }}_{n}^{n}(\tau_{+})e^{n}$$
for n=1,2,…,N and $\Delta_{+}\geq t\geq \Delta_{-}$. The above relationships are the same as satisfied by the FTNMs. In particular,
(112)$$\begin{array}{ll} {\underline{\psi }}^{n}(t) & ≐\underline{G}(t,\tau_{-}){\underline{\psi }}_{n}^{n}(\tau_{-})e^{n}≐\underline{G}(t,\tau_{-})({\tilde{\underline{\phi }}}_{n}^{n}(\tau_{-})e^{n},{\underline{\psi }}_{n}^{n}(\tau_{-})e^{n}){\tilde{\underline{\phi }}}_{n}^{n}(\tau_{-})e^{n}\\ \; & =({\tilde{\underline{\phi }}}_{n}^{n}(\tau_{-})e^{n},{\underline{\psi }}_{n}^{n}(\tau_{-})e^{n}){\tilde{\underline{\phi }}}_{}^{n}(t) \end{array}$$
where ${\tilde{\underline{\phi }}}_{}^{n}(t)={\underline{\phi }}_{}^{n}(t)/\left\|{\underline{\phi }}_{}^{n}(t)\right\|$. Thus, we can choose the orientations and amplitudes in Equation (112) so that ${\underline{\psi }}_{n}^{n}(t)≐{\underline{\phi }}_{n}^{n}(t).$

### 10.2. Transformation to the Original Phase-Space

The results in Equation (112) and following translate into near equivalence in the original x-space (Appendix C and Appendix D) for $\Delta_{+}\geq t\geq \Delta_{-}$:(113)$$\psi^{n}(t)=\Phi {\underline{\psi }}^{n}(t)≐\Phi {\underline{\phi }}^{n}(t)=\phi^{n}(t)$$
where $\Phi =\Phi (\tau_{-};[\tau_{+}-\tau_{-}])$ is the characteristic matrix.

Eichhorn et al. [76] note that the boundedness for all time of the nonsingular transformation matrices **Φ**^±1^, which depend on the reference trajectory and time, is sufficient for the convergence of the Lyapunov exponents (Appendix D, Equations (A73) to (A75)). The proof of the Oseledec Theorem 4 [72], which establishes the Lyapunov exponents and Oseledec subspaces, also uses a diagonalization of the propagator cocycle for dynamics in the subspaces. Oseledec used a homologous transformation of the propagator cocycle with the transformation matrices **Φ**^±1^ satisfying the Lyapunov condition (Appendix C, Equations (A54) and (A55)). The Lyapunov condition in Equation (A55), which means that **Φ**^±1^ have no asymptotic exponential growth, is slightly less stringent than the boundedness constraint in Equation (A73). With either condition, the relationship in Equation (113), between ***ψ***^*n*^(*t*) and ***ϕ***^*n*^(*t*) becomes equality for *τ*_±_→±∞. As noted, Δ_±_ = *γ*_±_*τ*_±_ and 1 > *γ*_±_ >0, so that Δ_±_→±∞ and we also have
$\Lambda_{r}^{n}(\tau_{+},\tau_{-})={\underline{\sum }}^{n}(\tau_{+},\tau_{-})\to L^{n}$.

For numerical calculations one can estimate the expansion of the CLVs by the FTNM exponents
(114)$$\Lambda^{n}(\tau_{+},\tau_{-})=\Lambda_{r}^{n}(\tau_{+},\tau_{-})+i\Lambda_{i}^{n}(\tau_{+},\tau_{-})$$
where the Lyapunov exponents, that are independent of finite time, are estimated by
(115)$$L^{n}≐{\underline{\sum }}^{n}(\tau_{+},\tau_{-})=\Lambda_{r}^{n}(\tau_{+},\tau_{-})$$
for n=1,2,…,N. Given the FTNMs between $\tau_{-}$ and $\tau_{+}$, the exponent in Equation (114) can also be calculated using
(116)$$\Lambda_{r}^{n}(\tau_{+},\tau_{-})=\frac{1}{\tau_{+}-\tau_{-}}\ln\left|\frac{{\underline{\phi }}_{n}^{n}(\tau_{+})}{{\underline{\phi }}_{n}^{n}(\tau_{-})}\right|$$
and
(117)Λin(τ+,τ−)=1τ+−τ−arctan[Im(ϕ_nn(τ+)ϕ_nn(τ−))Re(ϕ_nn(τ+)ϕ_nn(τ−))]One can also use Equations (116) and (117) with the replacements ${\underline{\phi }}_{n}^{n}\to {\underline{\psi }}_{n}^{n}$ and $\tau_{\pm }\to \Delta_{\pm }$ to estimate $L^{n}$ by $\Lambda_{r}^{n}(\Delta_{+},\Delta_{-})$ directly from the converged CLV in the interval $\Delta_{+}\geq t\geq \Delta_{-}$ in numerical calculations.

Appendix E presents a complementary way of analyzing the relationships between FTNMs and SVs in the long-time limits and OLVs and CLVs.

## 11. Calculation of Dynamical Vectors and Metric Entropy Production

Next, we briefly discuss some of the methods for efficient construction of dynamical instability vectors and propose a norm-independent finite-time measure of metric entropy production that characterizes instability and chaos.

### 11.1. Lyapunov Vectors

Efficient algorithms for the calculation of Lyapunov exponents and OLVs were initially proposed by Shimada and Nagashima [86] and Benetin et al. [87] and variations and improvements have been proposed in subsequent works [30,62,63]. The efficient numerical calculation of CLVs has proved to be more difficult. As discussed in Section 7, the relationships between norm-dependent OLVs and norm-independent CLVs were presented in the early works of Ruelle [61], Eckmann and Ruelle [62], Legras and Vautard [64] and Trevisan and Pancotti [65]. More efficient methods of constructing leading CLVs, from long-time SVs that approximate OLVs, were initially developed by Ginelli et al. [57] and Wolf and Samelson [56] and several variants have subsequently been developed [58,59,60]. For example, the method of Wolfe and Samelson [56] determines the leading n CLVs in terms of the leading n left OLVs, ${𝓾}^{n}(t)$, and n−1 right OLVs, ${𝓿}^{n}(t)$.

Wolfe and Samelson [56] emphasize important properties of CLVs including their norm-independence and the fact that if CLVs have been calculated at a particular time then in principle they can be obtained at all future times through propagation with the tangent linear equation and at earlier times with its inverse. In practice, however, as they also note in an example, it can be difficult to calculate the Lyapunov vectors with sufficient accuracy, even for low-order systems. In that case, CLVs can only be calculated by their method for a finite time before they tend towards the leading Lyapunov vector. Importantly, for aperiodic systems, the current methods of calculating CLVs from OLVs have largely been restricted to cases for which the Lyapunov exponents are distinct [56].

### 11.2. Degeneracy and Nondegeneracy

Unfortunately, for geophysical fluid dynamical systems where waves, such as Rossby waves, are prevalent the instability matrices generally result in complex conjugate eigenvalues [17,18,19,20,21,22,23,24,25,26,27] as do the propagators [27,49,50,79,80]. This has been the case for all the instability processes discussed in the Introduction from realistic storms to the large-scale low frequency disturbances with just a few stationary teleconnection patterns having real eigenvalues. For example, Floquet problems were considered by Frederiksen and Branstator [79,80] in which the propagator was calculated for the whole annual cycle for matrices of size 495×495. In the first study of the intra-annual variability of barotropic modes [79] there were just 55 distinct real Floquet eigenvalues and 440 complex eigenvalues in 220 distinct pairs. Thus, while the CLVs in this study are nondegenerate because the complex exponents are distinct, by far the majority of the real Lyapunov exponents have multiplicity 2 and the associated OLVs are degenerate. This is also the case in the Floquet studies of teleconnection patterns by Frederiksen and Branstator [80] using empirically determined propagators.

Frederiksen and Branstator [79] also considered the separable Floquet problem where the dynamical stability matrix $M(t)=c(t)M^{a}$ with $M^{\;a}$ a constant matrix characteristic of a typical annual average basic state and c(t)=c(t+T) represents the time varying strength of the matrix through the annual cycle T. Again, the eigenvalues and exponents of the annual dynamical matrix $M^{a}$, and the propagator derived from $M(t)=c(t)M^{a}$, mainly occur in distinct complex conjugate pairs with just a handful of distinct real exponents. One can of course also consider the corresponding separable aperiodic problem where c(t) is not periodic so that the real Lyapunov exponents are again mainly degenerate. This would also be the case with any of the three-dimensional instability matrices for synoptic disturbances from storms to teleconnection patterns discussed in the Introduction [17,18,19,20,21,22,23,24,25,26,27].

### 11.3. Finite-Time Normal Modes and Arnoldi Methods

As noted above, Frederiksen and Branstator [79,80] considered Floquet problems over a complete year. They generated sets of 12 monthly averaged dynamical matrices from which the propagator over a year was constructed as in Equation (8) with the short-time propagators calculated as in Equation (9) with a time step of half an hour. All the 495 eigenvalues and eigenvectors were calculated directly using LAPACK routines [92]. More generally, efficient algorithms have been developed for calculating just some of the leading fast-growing modes. Wei and Frederiksen [27], (Appendix), describe a very efficient algorithm for calculating some of the leading FTNMs that caters for aperiodic problems, as well as periodic problems. They used an Arnoldi iterative method [23,93,94,95,96] based on recycling perturbations [27] from an initial time $t_{0}$ to a final time $t_{f}$ with the tangent linear equation. The consequent FTNMs calculated at the initial time $t_{0}$ are then obtained at later times up to $t_{f}$ by forward propagation with the tangent linear equations.

Wei and Frederiksen [27] checked their results against calculations with the LAPACK routines [92] for systems with again 495 modes. However, the iterative method is applicable to complex high-dimensional systems since only a low-dimensional approximation to the propagator is needed, which is obtained from integrations of the tangent linear dynamical equations.

From the above examples it is evident that there are methods for computing FTNMs efficiently. In particular, the long-time FTNMs are computable for typical smooth systems with reasonably high numbers of degrees of freedom. Importantly, these methods are applicable to systems with complex FTNM exponents and associated degenerate SVs and OLVs.

### 11.4. Metric Entropy Production

Next, we consider measures of the chaotic nature of dynamical systems. The Kolmogorov-Sinai (KS) entropy production [8,9] may be approximated by Pesin’s formula [10] that expresses it as the sum of the positive Lyapunov exponents [11]:(118)$$\partial S_{KS}=\sum_{n=1}^{n_{P}}L^{n}.$$Here, $n_{P}$ is the largest index such that $L^{n_{P}}>0$. Another measure is the Hausdorff dimension of the phase-space which by the Kaplan-Yorke (KY) conjecture [7,97] is given by
(119)$$D_{KY}=n_{S}+\frac{\sum_{n=1}^{n_{S}}L^{n}}{\left|L^{n_{S}+1}\right|}$$
where $n_{S}$ is the largest index such that the sum $\sum_{n=1}^{n_{S}}L^{n}>0$.

As discussed in the Introduction, the local and finite-time growth rates of disturbances are more directly related to predictability than the long-time or global Lyapunov exponents [12,13,27,30,31,32,33,34,35,37,41]. Wei [12] proposed a local metric entropy (production) measure where the local Lyapunov exponent $L^{n}(t+\Delta t,t)$ replaces the global Lyapunov exponent in Equation (118). One can also generalize this to finite times with $L^{n}(t_{f},t_{0})$ replacing the global Lyapunov exponent. Unlike the global Lyapunov exponents there are several ways of defining the finite-time Lyapunov exponents [12,13,34,41]. They can be defined as the singular value exponents $\sum^{n}(t_{f},t_{0})$ in Equation (50) for the forward and backward problems or based on CLV norm expansion rates as in Equation (A69) of Appendix D. They can also be calculated using the standard method of Gram-Smidt orthogonalization [12,41,86,87].

As well as these different definitions and calculation methods there are two further issues with measures based on the finite-time Lyapunov growth rates. The first is that they are all norm dependent unlike the global Lyapunov exponents or FTNM growth rates. The second is that particularly for short times they express possible super exponential growth [41,50] while FTNM exponents represent exponential growth. The evidence from comprehensive weather forecast models [98,99], (see also [27,41]), is that errors conform closely with the 1982 Lorenz model [100] of exponential growth followed by nonlinear saturation. For these reasons, it is proposed that suitable finite-time generalizations of the Kolmogorov-Sinai entropy production and the Kaplan-Yorke conjectured Hausdorff dimension are the metric entropy production
(120)$$\partial S_{FT}=\sum_{n=1}^{n_{P}}\Lambda_{r}^{n}(t_{f},t_{0})$$
where $n_{P}$ is the largest index such that $\Lambda_{r}^{n_{P}}(t_{f},t_{0})>0$ and the dimension,
(121)$$D_{FT}=n_{S}+\frac{\sum_{n=1}^{n_{S}}\Lambda_{r}^{n}(t_{f},t_{0})}{\left|\Lambda_{r}^{n_{S}+1}(t_{f},t_{0})\right|}$$
where $n_{S}$ is the largest integer such that the sum $\sum_{n=1}^{n_{S}}\Lambda_{r}^{n}(t_{f},t_{0})>0$. Here, $\Lambda_{r}^{n}(t_{f},t_{0})$ is the FTNM growth rate defined in Equation (14). The proposed predictability time [12] is then
(122)$$T_{P}=\frac{D_{FT}}{\partial S_{FT}}.$$The finite-time metric entropy production in Equation (120) and dimension in Equation (121) of course become the KS entropy production and KY dimension respectively in the long time limits for the periodic systems of Section 9 as seen from Equation (92). Similarly, under the conditions on aperiodic systems described in Section 2 and Section 10, we expect the same convergence in the long time limit as seen from Equation (115).

## 12. Discussion and Conclusions

This study has examined the interrelated properties of dynamical instability vectors and exponents that have been, or may be, useful for understanding error growth and entropy production in geophysical fluid dynamical systems. A very practical application of such dynamical vectors is for ensemble perturbations in weather and seasonal climate prediction. A focus has been on the relationships between the norm independent CLVs and FTNMs and the norm dependent OLVs and SVs.

The Oseledec theorem [62,72,73,74,75] relates the long-time behaviour of SVs and singular value exponents to OLVs and Lyapunov exponents and in turn to CLVs. In the absence of a similar theorem between FTNMs and OLVs and CLVs their relationships to have not been at all clear. However, the Lyapunov exponents and Oseledec subspaces within which the Lyapunov vectors reside (see Appendix B) can be calculated in μ-almost any phase-space or norm [62,72] (see Appendix C and Appendix D). This result has been used here to examine the properties of dynamical instability vectors in FTNM-space. In FTNM-space the relationships between dynamical vectors simplify. In particular, the propagator between the initial time $t_{0}$ and final time $t_{f}$ becomes diagonal with the right, or initial, SVs and the left, or final, SVs as well as the initial and final FTNMs all proportional to the standard unit vectors. This means that at these critical times we can uniquely anchor FTNMs to SVs and use the Oseledec theorem to relate long-time FTNMs to OLVs and importantly to CLVs. Moreover, provided the FTNMs are nondegenerate, with generally complex FTNM exponents, the analysis can cater for associated degenerate SVs and OLVs that have multiple equal real exponents.

As noted in Section 2, the results of this study, particularly the equality of Lyapunov exponents with long-time instability growth rates of the propagator and the relationships between CLVs and FTNMs, depend on the properties of the dynamical system under consideration. Smooth ergodic dynamical systems are studied with bounded attractors for which the statistics are well defined [30,72,73,74,75,76] as these conditions underpin the proof of the multiplicative ergodic theorem by Oseledec for Lyapunov exponents and Oseledec subspaces. Oseledec [72] also assumed that the propagators are well defined and nonsingular. He proved his main theorem 4 by means of a diagonalization of the cocycle for dynamics in the subspaces ([72], p. 219). His Lyapunov conditions (Equation (A55)) on the homologous transformation matrix and its inverse also need to apply in the long-time limits to our characteristic matrices $\Phi^{\pm 1}$ of Equation (26). In our study, it is, in addition, assumed that the FTNMs are nondegenerate. If these conditions are not satisfied, at least for sufficiently large $T=t_{f}-t_{0}$, then it is possible to construct analytical counter examples where, for example, the stability spectrum becomes degenerate at some times, and the long-time stability growth rates and Lyapunov exponents are not the same [30]. However, in studies of systems with nondegenerate real stability exponents and Lyapunov exponents, associated with backward Lyapunov vectors, Goldhirsch et al. [30] present numerical evidence for their equality which they argue is the generic case.

Some of the main findings established in this study are:1.The covariant properties of FTNMs, $\phi^{n}(t)$, in the time interval $t_{f}\geq t\geq t_{0}$ are known and determined by $\phi^{n}(t)=G(t,t_{0})\phi^{n}(t_{0})$, as in Equation (19). We show that they also satisfy the eigenvalue problem $G(t+T,t_{0}+T)G(t_{0}+T,t)\phi^{n}(t)=\lambda^{n}\phi^{n}(t)$ in Equation (21) where $G(t+T,t_{0}+T)\equiv G(t,t_{0})$ by definition as noted in Equation (22). The propagator is in general discontinuous at $t_{0}+T=t_{f}$. In the case where it is continuous the FTNMs become Floquet vectors as noted in Equation (37). Indeed, in the efficient Arnoldi algorithm of Wei and Frederiksen [27], discussed in Section 11, the leading FTNMs are constructed by recycling perturbations from $t_{f}=t_{0}+T$ to $t_{0}$.2.In FTNM-space, the right, or initial, SVs ${\underline{v}}^{n}(t_{0})$ and left, or final, SVs ${\underline{u}}^{n}(t_{f})$, for n=1,2,…,N, are, like the FTNMs, ${\underline{\phi }}^{n}(t_{0})$ and ${\underline{\phi }}^{n}(t_{f})$, proportional to the unit vectors $e^{n}$ as shown in Equations (73) and (74). This is because the propagator between $t_{0}$ and $t_{f}$ in FTNM-space is normal, in fact diagonal (Equation (65)). A particular consequence is that in FTNM-space the singular value exponents are equal to the real part of the FTNM exponents: ${\underline{\sum }}^{n}(t_{f},t_{0})=\Lambda_{r}^{n}(t_{f},t_{0})$ as noted in Equation (76).3.The relationships, based on the Oseledec theorem [72], for the construction of OLVs from CLVs (Equations (53) and (54)) and importantly the determination of CLVs from OLVs (Equations (58) and (59)) and their approximations by long-time SVs in Section 7 greatly simplify at critical times in FTNM-space as shown in Section 9 and Section 10. This is because of the diagonalization of the propagator in result 2 above.4.In Appendix C, the propagator $G(t_{f},t_{0})$ in the original x-space and the propagator $\underline{G}(t_{f},t_{0})$ in FTNM-space have both been shown to be homologous to the stretching propagator $\tilde{\underline{G}}(t_{f},t_{0})$ with the transformation matrices subject to the Lyapunov condition in Equation (A55). This ensures that if the results of Oseledec’s [72] four theorems, including his multiplicative ergodic theorem 4, apply for a propagator in any of the phase-spaces then they hold for all these cohomologous propagator cocycles.5.For periodic systems, the results 3 and 4 above have been used in Section 9 to deduce the links of FTNMs to OLVs and CLVs, from Equations (53) to (59), and the equivalence of Floquet vectors and CLVs. Moreover, the Lyapunov exponents $L^{n}=\Lambda_{r}^{n}(t_{f},t_{0})={\underline{\sum }}^{n}(t_{f},t_{0})={\underline{\sum }}^{n}(t_{f},-\infty )={\underline{\sum }}^{n}(t_{0},-\infty )={\underline{\sum }}^{n}(\infty ,t_{0})={\underline{\sum }}^{n}(\infty ,t_{f})$, as given in Equation (92). This applies for systems with both real and complex Floquet exponents and nondegenerate and degenerate Lyapunov spectra.6.In the case of aperiodic systems, including with degenerate Lyapunov spectra, the results 3 and 4 above have been used in Section 10 to show that in the interval $\tau_{+}>\Delta_{+}\geq t\geq \Delta_{-}>\tau_{-}$, CLVs are closely approximated by FTNMs, $\psi^{n}(t)\approx \phi^{n}(t)$, for large $\left|\tau_{\pm }\right|$ with equality as $\tau_{\pm }\to \pm \infty$. Moreover, in FTNM-space the singular value exponent and FTNM growth rates are equal and approximate the Lyapunov exponent ${\underline{\sum }}^{n}(\tau_{+},\tau_{-})=\Lambda_{r}^{n}(\tau_{+},\tau_{-})≐L^{n}$ with equality as $\tau_{\pm }\to \pm \infty$.7.An alternative way of establishing the results in 5 and 6 is presented in Appendix E where FTNMs are orthogonalized using the Gram-Smidt method and the long-time limits of the orthonormal vectors are considered particularly in FTNM phase-space.8.Finite-time generalizations of the Kolmogorov-Sinai entropy production [8,9] and the Kaplan-Yorke conjectured Hausdorff dimension [97] have been proposed based on the FTNM exponents. The expressions, like the FTNMs, are norm-independent and may be reasonably accurate for ensembles of perturbations as noted in Section 11 although it is possible that intramodal and intermodal interference effects could contribute to individual perturbation growth [79,80].

In summary, CLVs have important theoretical characteristics for portraying the growth and nature of small amplitude perturbation evolution through the tangent linear equation. CLVs are norm independent [56,57] and so describe the same physics irrespective of whether the phase-space is described by the streamfunction, velocity or another commonly used variable or, for example, by empirical orthogonal functions or FTNMs. The CLVs are covariant with the tangent linear dynamics so if they are known at any one time can be obtained at any other time by propagation with the tangent linear propagator [56,57]. These properties of CLVs, for general aperiodic systems, also apply to Floquet vectors, and to FTNMs between the initial $t_{0}$ and final $t_{f}$ times for which they are defined. The finite time Lyapunov exponents, between $t_{0}$ and $t_{f}$, are however norm and formulation dependent (Section 11.4 and Appendix D) unlike the FTNM average growth rates, as discussed in the Introduction.

The numerical methods of Wolfe and Samelson [56] and Ginelli et al. [57], and subsequent refinements [58,59,60], have allowed more efficient calculation of the leading CLVs if the Lyapunov spectrum is nondegenerate [56]. The theoretical ideal of obtaining the future structures of CLVs from the current CLVs is in practice only possible for a finite time, even for relatively simple systems, before they tend towards the leading Lyapunov vector due to numerical errors [56].

For geophysical fluid dynamical systems with waves, such as Rossby waves, the Lyapunov spectrum is usually degenerate [79,80], and appropriate methods for the calculation of CLVs need to be explored. In this article we have shown that under suitable conditions the long-time FTNMs become CLVs and suggest that they be calculated as such. The convergence of any such approach will of course depend on the dimensionality of the system, the separation of the eigenvalues and the precision of the calculations. As a guide, the intermediate complexity atmospheric quasigeostrophic model studies of Reynolds and Errico [40] and Wei and Frederiksen [41] resolved some of the leading Lyapunov vectors on time scales of 30 to 40 days. From these investigations, and other works [27,49,50,79,80], it seems very likely that it should be possible to approximate leading CLVs, for simple, and even for intermediate complexity models, like barotropic [27,41,79] and baroclinic two- and three-level models [40,49,50]. Certainly, for these quasigeostrophic models that capture synoptic scale disturbances, calculating leading FTNMs over several months should be very achievable. We have also noted that for very smooth systems it is possible to calculate FTNMs and Floquet vectors for a whole year at reasonable resolution for barotropic systems [79,80]. Furthermore, efficient Arnoldi methods exist for the calculation of leading FTNMs by recycling perturbations with the tangent linear dynamical equations [27,41] and parallel algorithms can speed up the Arnoldi process [96] and may cater for more modes and higher dimensional systems. These methods should allow further exploration of the properties of FTNMs, including over long-time intervals, and determine their convergence to CLVs even when the Lyapunov spectrum is degenerate.

## Data Availability

Data sharing is not applicable since no new data were created or analyzed in this article.

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
