# Peer review of "Covariant Lyapunov Vectors and Finite-Time Normal Modes for Geophysical Fluid Dynamical Systems"

_entropy, 2023, doi:10.3390/e25020244_

Round 1
Reviewer 1 Report
Please find the attached file

Reviewer 2 Report
Summary: in this study, the author compares several different error growth analyses based on Lyapunov vectors, singular vectors, Floquet vectors, and finite-time normal mode eigenvectors. The presentation is mostly conceptual and informative with no new results or practical applications, as all types of above vectors including Lyapunov vectors or finite-normal mode eigenvectors have been previously explored and discussed in various studies. I was hoping that the authors could at least present a real-case study in which the difference among these vectors can be computed and compared against each other, but this expectation falls short. Despite no new result, I could see some value in this work in summarizing previous error growth vector concepts, which could be useful for future reference. With that, I have a few suggestions here so the author can take them into account before submitting the revision.
1. The notation used in this study is nonstandard and unnecessarily cumbersome, which makes it hard for readers to follow and compare to previous works. I would strongly suggest the author to revise your notation following the standard convention such as Yoden and Nomura (1993) so readers can appreciate your work more.
2. The amount of self-citation in this study is massive. Note that all of these concepts herein are not new, and have been previously explored. This work is more or less a review, and I do not see the actual contribution of the author’s claimed FTNM method for any real cases reported in this work. As such, I feel the number of self-citations should be confined, unless you want to present some actual results for real atmospheric systems that have not been previously shown;
3. I’d appreciate it if the author could outline how to obtain the propagator function $G(t_f,t_0)$ in Eq. (3) for full-physic numerical weather models such as WRF or GFS. The content presented herein is merely methodological, and how to apply it to real weather or climate systems is not sufficiently discussed.
4. The particular approximation of the propagator given by Eq. (9) is only valid when $\delta t$ is sufficiently small (relative to the time scale of a system). For fast-evolving weather systems such as mesoscale convective systems, (9) is no longer valid, yet these are the exact situations where error growth vectors are most needed. Even when (9) is applicable, how could you manage to get the explicit form for $M$ for practical problems or models?
5. It would be very helpful if you could add a table listing all current vectors and their properties, as well as related original references so readers can refer to them in the future.
Round 2
Reviewer 1 Report
Please see the attached file
